# REBEL: Reinforcement Learning via Regressing Relative Rewards

**Zhaolin Gao[1], Jonathan D. Chang[2]*, Wenhao Zhan[3], Owen Oertell[1], Gokul Swamy[4],**
**Kianté Brantley[5], Thorsten Joachims[1], J. Andrew Bagnell[4,6], Jason D. Lee[3], Wen Sun[1]**

[1] Cornell University, [2] Databricks Mosaic Research, [3] Princeton University,
[4] Carnegie Mellon University, [5] Harvard University, [6] Aurora Innovation

## Abstract

While originally developed for continuous control problems, Proximal Policy Optimization (PPO) has emerged as the work-horse of a variety of reinforcement learning (RL) applications, including the fine-tuning of generative models. Unfortunately, PPO requires multiple heuristics to enable stable convergence (e.g. value networks, clipping), and is notorious for its sensitivity to the precise implementation of these components. In response, we take a step back and ask what a *minimalist* RL algorithm for the era of generative models would look like. We propose REBEL, an algorithm that cleanly reduces the problem of policy optimization to regressing the *relative reward* between two completions to a prompt in terms of the policy, enabling strikingly lightweight implementation. In theory, we prove that fundamental RL algorithms like Natural Policy Gradient can be seen as variants of REBEL, which allows us to match the strongest known theoretical guarantees in terms of convergence and sample complexity in the RL literature. REBEL can also cleanly incorporate offline data and be extended to handle the intransitive preferences we frequently see in practice. Empirically, we find that REBEL provides a unified approach to language modeling and image generation with stronger or similar performance as PPO and DPO, all while being simpler to implement and more computationally efficient than PPO. When fine-tuning Llama-3-8B-Instruct, REBEL achieves strong performance in AlpacaEval 2.0, MT-Bench, and Open LLM Leaderboard. Implementation of REBEL can be found at https://github.com/ZhaolinGao/REBEL, and models trained by REBEL can be found at https://huggingface.co/Cornell-AGI.

## 1 Introduction

The generality of the reinforcement learning (RL) paradigm is striking: from continuous control problems (Kalashnikov et al., 2018) to, more recently, the fine-tuning of generative models (Stiennon et al., 2022; Ouyang et al., 2022), RL has enabled concrete progress across a variety of decision-making tasks. Specifically, when it comes to fine-tuning generative models, Proximal Policy Optimization (PPO, Schulman et al. (2017)) has emerged as the de-facto RL algorithm of choice, from language models (LLMs) (Ziegler et al., 2020; Stiennon et al., 2022; Ouyang et al., 2022; Touvron et al., 2023) to generative image models (Black et al., 2023; Fan et al., 2024; Oertell et al., 2024).

---

*Work done at Cornell
{zg292,ojo2,tj36,ws455}@cornell.edu, j.chang@databricks.com, kdbrantley@g.harvard.edu,
{wenhao.zhan,jasonlee}@princeton.edu, {gswamy,bagnell2}@andrew.cmu.edu

38th Conference on Neural Information Processing Systems (NeurIPS 2024).

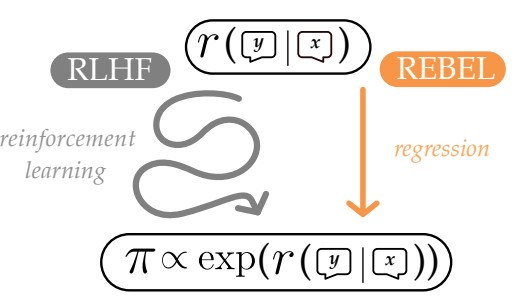 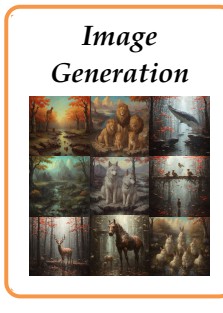 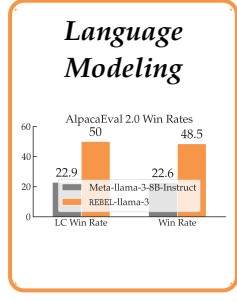

Figure 1: We present REBEL: a simple and scalable RL algorithm that performs policy optimization via *iteratively regressing the difference in rewards in terms of the policy*, allowing us to eliminate much of the complexity (e.g. value functions, clipping) of algorithms like PPO (Schulman et al., 2017). We apply REBEL to problems in both image generation and language modeling and find that despite its conceptual and implementation-level simplicity, REBEL is able to match or sometimes outperform the performance of PPO while out-performing purely offline techniques like DPO (Rafailov et al., 2023). REBEL also achieves strong performance on common benchmarks such as AlpacaEval when fine-tuning a Llama-3-8B model.

If we take a step back however, it is odd that we are using an algorithm designed for optimizing two-layer networks for continuous control tasks from scratch, even though we are now fine-tuning generative models with billions of parameters. In the continuous control setting, the randomly initialized neural networks and the possible stochasticity in the dynamics necessitate variance reduction through a learned value function as a baseline (Schulman et al., 2015), while clipping updates is important to limit distribution shift from iteration to iteration (Kakade and Langford, 2002). This means that when applied to generative model fine-tuning, we need to store four models in memory simultaneously (the policy, the reference policy, the critic, and the reward model), each with billions of parameters. Furthermore, we often add a KL regularization to the base model for fine-tuning, making explicit clipping unnecessary nor advisable, as pointed out by Ahmadian et al. (2024). Even outside of the generative modeling context, PPO is notorious for the wide range of performances measured, with differences being attributed to seemingly inconsequential implementation details (Henderson et al., 2019; Engstrom et al., 2020). This begs the question: *are there simpler algorithms that better scale to modern RL applications?*

Our answer is REBEL: an algorithm that *reduces the problem of RL to solving a sequence of squared loss regression problems* on iteratively collected datasets. Each regression problem directly uses the policy to predict the difference in rewards. This allows us to eliminate the complexity of using value functions, avoids heuristics like clipping, and scales easily to problems in both language modeling and image generation. Our key insight is that *a regressor that can predict the difference in rewards between trajectories in a dataset implicitly captures an improved policy.*

Rather than being a mere heuristic, REBEL comes with strong guarantees in theory and can be seen as a strict generalization of classical techniques (e.g., NPG) in reinforcement learning. Furthermore, REBEL cleanly incorporates offline datasets when available, can be extended to robustly handle intransitive preferences (Swamy et al., 2024), empirically out-performs techniques like PPO and DPO (Rafailov et al., 2023) in language generation, and has a faster convergence with a similar asymptotic performance in image generation. When fine-tuning a Llama-3-8B model, REBEL also demonstrates very competitive performance on AlpacaEval 2.0, MT-bench, Open LLM Leaderboard, and Arena Hard simultaneously. We begin by formalizing the preference fine-tuning setup before deriving our core algorithmic technique.

## 2 REBEL: REgression to RElative REward Based RL

We consider the contextual bandit formulation (Langford and Zhang, 2007) of RL which has been used to formalize the generation process of models like LLMs (Rafailov et al., 2023; Ramamurthy et al., 2022; Chang et al., 2023) and Diffusion Models (Black et al., 2023; Fan et al., 2024; Oertell et al., 2024) due to the determinism of the transitions. More explicitly, in the deterministic transition

---

**Algorithm 1** REgression to RElative REward Based RL (REBEL)

---

1: **Input**: Reward $r$, policy class $\Pi = \{\pi_\theta : \theta \in \Theta\}$, base distribution $\mu$, learning rate $\eta$
2: Initialize policy $\pi_{\theta_0}$.
3: **for** $t = 0$ to $T - 1$ **do**
4:     `// Base distribution` $\mu$ `can either be an offline dataset or` $\pi_t$`.`
5:     Collect dataset $\mathcal{D}_t = \{x, y, y'\}$ where $x \sim \rho, y \sim \pi_t(\cdot|x), y' \sim \mu(\cdot|x)$.
6:     Solve square loss regression problem:

$$\theta_{t+1} = \underset{\theta \in \Theta}{\operatorname{argmin}} \sum_{(x,y,y') \in \mathcal{D}_t} \left( \frac{1}{\eta} \left( \ln \frac{\pi_\theta(y|x)}{\pi_{\theta_t}(y|x)} - \ln \frac{\pi_\theta(y'|x)}{\pi_{\theta_t}(y'|x)} \right) - (r(x,y) - r(x,y')) \right)^2.$$
(1)

7: **end for**

---

setting, explicit states are not required as they are isomorphic to the sequence of actions. Furthermore, the entire sequence of actions can be considered as a single "arm" in a bandit problem with an action space that scales exponentially in size with the horizon of the problem.

We denote by $(x, y)$ a (prompt, response) pair, where $x \in \mathcal{X}$ is the prompt and $y \in \mathcal{Y}$ is the response (e.g. a sequence of tokens, or in general a sequence of actions). We assume access to a reward function $r(x, y)$ from which we can query for reward signals (the exact form of $r$ does not need to be known). Querying $r$ at $(x, y)$ will return a scalar $r(x, y)$, measuring the quality of the prompt completion. Such a reward function could be a pre-defined metric (e.g., Rouge score against human responses) or a learned model from an offline human demonstration or preference data (e.g. the RLHF paradigm (Christiano et al., 2017; Ziegler et al., 2020)), as we focus on in our experiments.

Denote by $\pi \in \mathcal{X} \mapsto \Delta(Y)$ a policy (e.g. an LLM) that maps from a prompt $x$ to a distribution over the response space $\mathcal{Y}$. We use $\rho$ to denote the distribution over prompts (i.e. initial states / contexts) $x$ and $\pi_\theta(y|x)$ to denote a policy with parameter $\theta$. At times, we interchangeably use $\pi_t$ and $\pi_{\theta_t}$ when it is clear from the context. We emphasize that while we focus on the bandit formulation for notational simplicity, the algorithms proposed here can be applied to *any* deterministic MDP where $x$ is the initial state and the trajectory $y$ consists of the sequence of actions.

At each iteration of all algorithms, our goal will be to solve the following KL-constrained RL problem:

$$\pi_{t+1} = \underset{\pi \in \Pi}{\operatorname{argmax}} \, \mathbb{E}_{x, y \sim \pi(\cdot|x)} r(x, y) - \frac{1}{\eta} \mathbb{E}_x \text{KL} \left( \pi(\cdot|x) || \pi_t(\cdot|x) \right).$$
(2)

Intuitively, this can be thought of asking for the optimizer to fine-tune the policy $\pi_{t+1}$ according to $r$ while staying close in terms of action distribution to some baseline policy $\pi_t$.

### 2.1 Deriving REBEL: REgression to RElative REward Based RL

From Ziebart et al. (2008), we know that there exists a closed-form solution to the above *minimum relative entropy* problem (Eq. 2, Grünwald and Dawid (2004)):

$$\forall x, y : \pi_{t+1}(y|x) = \frac{\pi_t(y|x) \exp(\eta r(x, y))}{Z(x)}; \; Z(x) = \sum_y \pi_t(y|x) \exp(\eta r(x, y)).$$
(3)

As observed by Rafailov et al. (2023), we can invert Eq. 3 and write the reward in terms of the policy:

$$\forall x, y : r(x, y) = \frac{1}{\eta} \left( \ln(Z(x)) + \ln \left( \frac{\pi_{t+1}(y|x)}{\pi_t(y|x)} \right) \right).$$
(4)

As soon as $\mathcal{X}$ and $\mathcal{Y}$ become large, we can no longer guarantee the above expression holds exactly at all $(x, y)$ and therefore need to turn our attention to choosing a policy such that Eq. 4 is approximately true. We propose using a simple *square loss* objective between the two sides of Eq. 4 to measure the goodness of a policy, i.e. reducing RL to a regression problem: $\left( r(x, y) - \frac{1}{\eta} \left( \ln(Z(x)) + \ln \left( \frac{\pi_{t+1}(y|x)}{\pi_t(y|x)} \right) \right) \right)^2$. Unfortunately, this loss function includes the *partition function* $Z(x)$, which can be challenging to approximate over large input / output domains.

However, observe that $Z(x)$ only depends on $x$ and not $y$. Thus, if we have access to *paired samples*, i.e. $(x, y)$ and $(x, y')$, we can instead regress the *difference in rewards* to eliminate this term:

$$\left( (r(x, y) - r(x, y')) - \frac{1}{\eta} \left( \ln \left( \frac{\pi_{t+1}(y|x)}{\pi_t(y|x)} \right) - \ln \left( \frac{\pi_{t+1}(y'|x)}{\pi_t(y'|x)} \right) \right) \right)^2. \tag{5}$$

Of course, we need to evaluate this loss function on some distribution of samples. In particular, we propose using an on-policy dataset $\mathcal{D}_t = \{x, y, y'\}$ with $x \sim \rho, y \sim \pi_t(\cdot|x), y' \sim \mu(\cdot|x)$, where $\mu$ is some *base distribution*. The base distribution $\mu$ can either be a fixed offline dataset (e.g. the instruction fine-tuning dataset) or $\pi_t$ itself. Thus, the choice of base distribution $\mu$ determines whether REBEL is hybrid or fully online. Putting it all together, we arrive at our core REBEL objective in Eq. 1. Critically, observe that if we were able to perfectly solve this regression problem, we would indeed recover the optimal solution to the KL-constrained RL problem we outlined in Eq. 2.

## 3 Understanding REBEL as an Adaptive Policy Gradient

In this section, we interpret REBEL as an adaptive policy gradient method to illuminate the relationship to past techniques. We start by introducing algorithms such as Mirror Descent, NPG, and PPO, followed by illustrating why REBEL addresses the limitations of these past algorithms. For concision, we postpone an in-depth discussion of related work to Appendix A.

### 3.1 Adaptive Gradient Algorithms for Policy Optimization

**Mirror Descent.** If $\mathcal{X}$ and $\mathcal{Y}$ are small discrete spaces, we can use the closed-form expression for the minimum relative entropy problem (Eq. 3). This is equivalent to the classic Mirror Descent (MD) algorithm with KL as the Bregman divergence. Both NPG and PPO are approximations of MD.

**Natural Policy Gradient.** When $\mathcal{Y}$ and $\mathcal{X}$ are large, we use a parameterized policy denoted as $\pi_\theta$ with parameter $\theta$. Natural Policy Gradient (NPG, Kakade (2001)) approximates the KL in Equation 2 via its second-order Taylor expansion, whose Hessian is known as the Fisher Information Matrix (FIM, Bagnell and Schneider (2003)), $F_t$, i.e. $F_t = \mathbb{E}_{x, y \sim \pi_{\theta_t}(\cdot|x)} \left[ \nabla \ln \pi_{\theta_t}(y|x) \nabla \ln \pi_{\theta_t}(y|x)^\top \right]$. Thus, $\mathbb{E}_x \mathrm{KL}(\pi_\theta(\cdot|x) || \pi_{\theta_t}(\cdot|x)) \approx (\theta - \theta_t)^\top F_t (\theta - \theta_t)$. The NPG update can be formulated as:

$$\theta_{t+1} = \theta_t + \eta F_t^\dagger \left( \mathbb{E}_{x, y \sim \pi_{\theta_t}(\cdot|x)} \nabla \ln \pi_{\theta_t}(y|x) r(x, y) \right) \tag{6}$$

where $F_t^\dagger$ is pseudo-inverse of $F_t$. As mentioned above, this update procedure can be understood as performing gradient updates in the local geometry induced by the Fisher information matrix, which ensures that we are taking small steps in *policy space* rather than in *parameter space*. NPG, unfortunately, does not scale to modern settings due to need of inverting the FIM at each iteration.

**Proximal Policy Optimization.** Proximal Policy Optimization (PPO, Schulman et al. (2017)) takes a more direct route than NPG and uses clipped updates

$$\theta_{t+1} := \underset{\theta}{\mathrm{argmax}} \, \mathbb{E}_{x, y \sim \pi_{\theta_t}(\cdot|x)} \mathrm{clip} \left( \frac{\pi_\theta(y|x)}{\pi_{\theta_t}(y|x)}; 1 - \epsilon, 1 + \epsilon \right) r(x, y). \tag{7}$$

While the clipping operator can set the gradient to be zero at samples $(x, y)$ where $\pi_{\theta_{t+1}}(y|x)$ is much larger or smaller than $\pi_{\theta_t}(y|x)$, it cannot actually guarantee $\pi_{\theta_{t+1}}$ staying close to $\pi_{\theta_t}$, a phenomenon empirically observed in prior work (Hsu et al., 2020). Furthermore, hard clipping is not adaptive – it treats all $(x, y)$ equally and clips whenever the ratio is outside of a fixed range. In contrast, constraining the KL divergence to the prior policy allows one to vary the ratio $\pi(y|x)/\pi_t(y|x)$ at different $(x, y)$, as long as the total KL divergence across the state space is small. Lastly, clipping reduces the effective size of a batch of training examples and thus wastes training samples.

### 3.2 Connections between REBEL and MD / NPG

**Exact REBEL is Mirror Descent.** First, to build intuition, we interpret our algorithm's behavior under the assumption that the least square regression optimization returns the exact Bayes Optimal solution (i.e., our learned predictor achieves zero prediction error everywhere):

$$\forall x, y, y' : \quad \frac{1}{\eta} \left( \ln \frac{\pi_{\theta_{t+1}}(y|x)}{\pi_{\theta_t}(y|x)} - \ln \frac{\pi_{\theta_{t+1}}(y'|x)}{\pi_{\theta_t}(y'|x)} \right) = r(x, y) - r(x, y') \tag{8}$$

Conditioned on Eq. 8 being true, a few lines of algebraic manipulation reveal that there must exist a function $c(x)$ which is independent of $y$, such that $\forall x, y : \frac{1}{\eta} \ln \frac{\pi_{\theta_{t+1}}(y|x)}{\pi_{\theta_t}(y|x)} = r(x,y) + c(x)$. Taking an exp on both sides and re-arrange terms, we get $\forall x, y : \pi_{\theta_{t+1}}(y|x) \propto \pi_{\theta_t}(y|x) \exp{(\eta r(x,y))}$. In other words, under the strong assumption that least square regression returns a point-wise accurate estimator (i.e., Eq. 8), we see the REBEL recovers the exact MD update, which gives it *(a)* a fast $1/T$ convergence rate (Shani et al., 2020; Agarwal et al., 2021), *(b)* conservativity, i.e., $\max_x \mathrm{KL}(\pi_{t+1}(\cdot|x)||\pi_t(\cdot|x))$ is bounded as long as $\max_{x,y} |r(x,y)|$ is bounded, and *(c)* monotonic policy improvement via the NPG standard analysis (Agarwal et al., 2021).

**NPG is Approximate** REBEL **with Gauss-Newton Updates.** We provide another interpretation of REBEL by showing that NPG (Eq. 6) can be understood as a special case of REBEL where the least square problem in Eq. 1 is approximately solved via a single iteration of the Gauss-Newton algorithm. We start by approximating our predictor $\frac{1}{\eta} \ln \pi_\theta(y|x)/\pi_{\theta_t}(y|x)$ by its first order Taylor expansion at $\theta_t$: $\frac{1}{\eta} (\ln \pi_\theta(y|x) - \ln \pi_{\theta_t}(y|x)) \approx \frac{1}{\eta} \nabla_\theta \ln \pi_{\theta_t}(y|x)^\top (\theta - \theta_t)$, where $\approx$ indicates that we ignore higher order terms in the expansion. Define $\delta := \theta - \theta_t$ and replace $\frac{1}{\eta} (\ln \pi_\theta(y|x) - \ln \pi_{\theta_t}(y|x))$ by its first order approximation in Eq. 1. Then, we have :

$$\min_\delta \mathbb{E}_{x\sim\rho, y\sim\pi_{\theta_t}(\cdot|x), y'\sim\mu(\cdot|x)} \left( \frac{1}{\eta} \left(\nabla_\theta \ln \pi_{\theta_t}(y|x) - \nabla_\theta \ln \pi_{\theta_t}(y'|x)\right)^\top \delta - (r(x,y) - r(x,y')) \right)^2 \tag{9}$$

Further simplifying notation, we denote the uniform mixture of $\pi_t$ and $\mu$ as $\pi_{mix}(\cdot|x) := (\pi_t(\cdot|x) + \mu(\cdot|x))/2$ and the Fisher information matrix $F_t$ averaged under said mixture as $F_t = \mathbb{E}_{x\sim\rho, y\sim\pi_{mix}(\cdot|x)} \left[ \nabla_\theta \ln \pi_{\theta_t}(y|x) \left(\nabla_\theta \ln \pi_{\theta_t}(y|x)\right)^\top \right]$. Solving the above least squares problem to obtain a minimum norm solution, we have the following result.

**Claim 1.** *The minimum norm minimizer $\delta^\star$ of the least squares problem in Eq. 9 recovers an advantage-based NPG update:* $\delta^\star := \eta F_t^\dagger \left( \mathbb{E}_{x\sim\rho, y\sim\pi_{mix}(\cdot|x)} \nabla_\theta \ln \pi_{\theta_t}(y|x) [A^{\pi_t}(x,y)] \right)$ *where $F_t^\dagger$ is pseudo-inverse of $F_t$, and the* advantage *is defined as $A^{\pi_t}(x,y) := r(x,y) - \mathbb{E}_{y'\sim\pi_t(\cdot|x)} r(x,y)$.*

The proof of this claim is deferred to Appendix B.

**The implicit variance reduction effect of** REBEL We show that regressing to relative rewards has a variance reduction effect by extending the previous derivation on REBEL with Gauss-Newton update to the setting of finite data $\mathcal{D} = \{x_n, y_n, y'_n\}_{n=1}^N$. Denote the unbiased estimate of the Fisher information matrix as $\hat{F}_t = \frac{1}{N} \sum_{n=1}^N \left[ \nabla_\theta \ln \pi_{\theta_t}(y_n|x_n) \left(\nabla_\theta \ln \pi_{\theta_t}(y_n|x_n)\right)^\top \right]$ and have the following claim.

**Claim 2.** *The minimum norm minimizer $\delta^\star$ in Eq. 9 under finite setting has the form $\delta^\star := \eta \hat{F}_t^\dagger \frac{1}{2N} \sum_n \left( \nabla \ln \pi_{\theta_t}(y_n|x_n)(r(x_n, y_n) - r(x_n, y'_n)) + \nabla \ln \pi_{\theta_t}(y'_n|x_n)(r(x_n, y'_n) - r(x_n, y_n)) \right)$ where $\hat{F}_t^\dagger$ is pseudo-inverse of $\hat{F}_t$.*

The proof of this claim is deferred to Appendix C. Looking at the gradient formulation $\nabla \ln \pi_{\theta_t}(y_n|x_n) (r(x_n, y_n) - r(x_n, y'_n))$ in $\delta^\star$, we see that $r(x_n, y'_n)$ serves as a baseline for variance reduction. Interestingly, this gradient formulation is similar to RLOO (REINFORCE with leave-one-out) (Kool et al., 2019). However, different from RLOO, we pre-condition this variance reduced policy gradient formulation via the Fisher information matrix, leading to better performance.

**A** REBEL *With a Cause.* Our algorithm REBEL addresses the limitations of NPG (scalability) and PPO (lack of conservativity or adaptivity) from above. First, unlike NPG, it does not rely on the Fisher Information Matrix at all and can easily scale to modern LLM and image generation applications, yet can be interpreted as a *generalization* of NPG. Second, in contrast to PPO, it doesn't have unjustified heuristics and thus enjoys strong convergence and regret guarantees just like NPG. Building on Swamy et al. (2024), we also show how to extend REBEL to preference-based settings without assuming transitivity in Appendix D.

## 4 Theoretical Analysis

In the previous section, we interpret REBEL as exact MD and show its convergence by assuming that least square regression always returns a predictor that is accurate *everywhere*. While such

an explanation is simple and has also been used in prior work (Calandriello et al., 2024; Rosset et al., 2024), point-wise out-of-distribution generalization is an extremely strong condition and is significantly beyond what a standard supervised learning method can promise. In this section, we substantially relax this condition via a reduction-based analysis: ***As long as we can solve the regression problems well in an in-distribution manner, REBEL can compete against any policy covered by the training data distributions***. Formally, we assume the following generalization condition holds on the regressors we find.

**Assumption 1** (Regression generalization bounds). *Over $T$ iterations, assume that for all $t$, we have the following for some $\epsilon$:*

$$\mathbb{E}_{x \sim \rho, y \sim \pi_t(\cdot|x), y' \sim \mu(\cdot|x)} \left( \frac{1}{\eta} \left( \ln \frac{\pi_{\theta_{t+1}}(y|x)}{\pi_{\theta_t}(y|x)} - \ln \frac{\pi_{\theta_{t+1}}(y'|x)}{\pi_{\theta_t}(y'|x)} \right) - (r(x,y) - r(x,y')) \right)^2 \le \epsilon,$$

Detailed justifications for this assumption are provided in Appendix E.

**Data Coverage**. Recall that the base distribution $\mu$ can be some behavior policy, which in RLHF can be a human labeler, a supervised fine-tuned policy (SFT), or just the current learned policy (i.e., on-policy). Given a test policy $\pi$, we denote by $C_{\mu \to \pi}$ the concentrability coefficient, i.e.

$$C_{\mu \to \pi} = \max_{x,y} \frac{\pi(y|x)}{\mu(y|x)}. \tag{10}$$

We say $\mu$ *covers* $\pi$ if $C_{\mu \to \pi} < +\infty$. Our goal is to bound the regret between our learned policies and an arbitrary comparator $\pi^*$ (e.g. the optimal policy if it is covered by $\mu$) using $\epsilon$ and the concentrability coefficient defined in Eq. 10. The following theorem formally states the regret bound of our algorithm.

**Theorem 1.** *Under Assumption 1, after $T$ many iterations, with a proper learning rate $\eta$, among the learned policies $\pi_1, \ldots, \pi_T$, there must exist a policy $\hat{\pi}$, such that:*

$$\forall \pi^* : \ \mathbb{E}_{x \sim \rho, y \sim \pi^*(\cdot|x)} r(x,y) - \mathbb{E}_{x \sim \rho, y \sim \hat{\pi}(\cdot|x)} r(x,y) \le O \left( \sqrt{\frac{1}{T}} + \sqrt{C_{\mu \to \pi^*} \epsilon} \right).$$

The above theorem shows a ***reduction from RL to supervised learning*** — as long as supervised learning works (i.e., $\epsilon$ is small), then REBEL can compete against any policy $\pi^*$ that is covered by the base data distribution $\mu$. In the regret bound, the $1/\sqrt{T}$ comes from Mirror Descent style update, and $C_{\mu \to \pi^*} \epsilon$ captures the cost of distribution shift: we train our regressors under distribution $\pi_t$ and $\mu$, but we want the learned regressor to predict well under $\pi^*$. Similar to the NPG analysis from Agarwal et al. (2021), we now have a slower convergence rate $1/\sqrt{T}$, due to the fact that we have approximation error from learning. Such an agnostic regret bound — being able to compete against any policy that is covered by training distributions — is the **strongest type of agnostic learning results known in the RL literature**, matching the best of what has appeared in prior policy optimization work including PSDP (Bagnell et al., 2003), CPI (Kakade and Langford, 2002), NPG (Agarwal et al., 2021), and PC-PG (Agarwal et al., 2020). While in this work we use the simplest and most intuitive definition of coverage – the density ratio-based definition in Eq. 10 – extension to more general ones such as transfer error (Agarwal et al., 2020, 2021) or concentrability coefficients that incorporate the function class (e.g., Song et al. (2023)) is straightforward. We defer the proof of the above theorem and the detailed constants that we omitted in the $O$ notation to Appendix F. We include an extension of the above analysis to the general preference setting in Appendix G.

**Remark 1** (Discussion on the size of the response space $|\mathcal{Y}|$ and other design choices of the sampling distributions). *In REBEL, when sampling a pair $(y, y')$, we in default sample $y \sim \pi_t$, i.e., we make sure at least one of them is an on-policy sample. This is to make sure that the training distribution at iteration $t$ covers $\pi_t$, which plays an essential role in avoiding a polynomial dependency on the size of the action space $|\mathcal{Y}|$.[2] On the other hand, as long as we have some off-policy distribution $\nu_t$ that covers $\pi_t$ for all $t$, we can use it to sample $y$ and pay an additional concentrability coefficient*

---

[2]The sample complexity of the Q-NPG algorithm presented from Agarwal et al. (2019) has a polynomial dependence on the size of the action space since it samples actions uniform randomly in order to cover both $\pi_t$ and $\pi^*$. REBEL leverages that we can reset from the same context $x$, and thus directly draw two samples per context – one from $\pi_t$ and one from $\mu$, to cover $\pi_t$ and $\pi^*$ simultaneously.

| Model size | Algorithm | Winrate (↑) | RM Score (↑) | $\mathrm{KL}(\pi\|\pi_{ref})$ (↓) |
|---|---|---|---|---|
| 1.4B | SFT | 24.9 (±2.73) | -0.51 (±0.05) | - |
| | DPO | 42.7 (±1.79) | 0.10 (±0.02) | 29.6 (±0.63) |
| | Iterative DPO | 47.2 (±1.34) | 1.73 (±0.05) | 29.7 (±0.57) |
| | PPO | 51.7 (±1.42) | 1.74 (±0.04) | **29.3** (±0.61) |
| | REBEL | **55.1** (±1.35) | **1.84** (±0.04) | 32.6 (±0.59) |
| 2.8B | SFT | 28.2 (±2.31) | -0.38 (±0.06) | - |
| | DPO | 53.7 (±1.63) | 2.40 (±0.02) | 64.3 (±1.25) |
| | Iterative DPO | 63.1 (±1.41) | 2.37 (±0.03) | 28.1 (±0.51) |
| | PPO | 67.4 (±1.30) | 2.37 (±0.03) | **27.2** (±0.55) |
| | REBEL | **70.2** (±1.32) | **2.44** (±0.02) | 29.0 (±0.60) |

| Model size | Algorithm | Winrate (↑) |
|---|---|---|
| 6.9B | SFT | 45.2 (±2.49) |
| | DPO | 68.4 (±2.01) |
| | REINFORCE | 70.7[*] |
| | PPO | 77.6[‡] |
| | RLOO ($k=2$) | 74.2[*] |
| | RLOO ($k=4$) | 77.9[*] |
| | REBEL | **78.1** (±1.74) |

[*] directly obtained from Ahmadian et al. (2024)
[‡] directly obtained from Huang et al. (2024)

Table 1: **Results on *TL;DR* Summarization.** Results are averaged over three seed and the standard deviations across seeds are in parentheses. The best-performing method for each size and metric is highlighted in bold and the second best is underlined. REBEL outperforms all baselines on winrate.

$\max_{x,y,t} \pi_t(y|x)/\nu_t(y|x)$ *in the final bound. In experiments, we test the combination of the best-of-N of $\pi_t$ as the base distribution $\mu$ and the worst-of-N of $\pi_t$ as the $\nu_t$. Setting $\mu$ to be the best-of-N of $\pi_t$ makes $\mu$ cover higher quality comparator policies. Selecting $\nu_t$ as the worst-of-N of $\pi_t$ still ensures coverage to $\pi_t$ while at the same time increasing the reward gap $r(x,y) - r(x,y')$, which we find is helpful experimentally.*

## 5 Experiments

Our implementation of REBEL closely follows the psuedocode in Algorithm 1. In each iteration, REBEL collects a dataset $\mathcal{D}_t = \{x, y, y'\}$, where $x \sim \rho, y \sim \pi_t(\cdot|x), y' \sim \mu(\cdot|x)$. Subsequently, REBEL optimizes the least squares regression problem in Eq. 1 through gradient descent with AdamW (Loshchilov and Hutter, 2017). We choose $\mu = \pi_t$ such that both $y$ and $y'$ are generated by the current policy. We empirically assess REBEL's performance on both natural language generation and text-guided image generation. Additional experiment details are in Appendix H.

### 5.1 Summarization

**Task.** We use the *TL;DR* dataset (Stiennon et al., 2020) where $x$ is a forum post from Reddit and $y$ is a summary generated by the policy. The dataset comprises human reference summaries and preference data. We compare REBEL with baseline RL algorithms, REINFORCE (Williams, 1992) and its multi-sample extension, REINFORCE Leave-One-Out (RLOO) (Kool et al., 2019), PPO (Schulman et al., 2017), Direct Preference Optimization (DPO) (Rafailov et al., 2023), and Iterative DPO (Guo et al., 2024). Our implementation of Iterative DPO replaces our square regression objective with the DPO objective where the binary preference labels are obtained based on the reward difference. The implementation detail of the baseline methods is provided in Appendix H.1.3. Following prior work (Stiennon et al., 2020; Rafailov et al., 2023; Ahmadian et al., 2024), we train DPO on the preference dataset, while conducting online RL (RLOO, PPO, Iterative DPO, REBEL) on the human reference dataset. We include results with three different model sizes: 1.4B, 2.8B, and 6.9B based on the pre-trained models from Pythia (Biderman et al., 2023). Each model is trained from a supervised fine-tuned (SFT) model using a reward model (RM) of the same size.

**Evaluation.** We evaluate each method by its balance between reward model score and KL-divergence with the SFT policy, testing the effectiveness of the algorithm in optimizing the regularized RL objective. To evaluate the quality of the generation, we compute the winrate (Rafailov et al., 2023) against human references using GPT4 (OpenAI, 2023). The winrate is computed from a randomly sampled subset (10%) of the test set with 600 samples. We report the average results over three seeds.

**Quality Analysis.** Table 1 presents a comparison between REBEL and baseline methods. Notably, REBEL outperforms all the baselines on RM score with 1.4B and 2.8B parameters with a slightly larger KL than PPO. In addition, REBEL achieves the highest winrate under GPT4 when evaluated against human references, indicating the benefit of regressing the relative rewards. An ablation analysis on parameter $\eta$ is in Appendix J and the trade-off between the reward model score and KL-divergence is discussed in Appendix K.

**Runtime & Memory Analysis.** We analyze the runtime and peak memory usage for 2.8B models with REINFORCE, RLOO, PPO, DPO, Iterative DPO, and REBEL. The runtime includes both the generation time and the time required for policy updates. Both runtime and peak memory usage are measured on A6000 GPUs using the same hyperparameters detailed in Appendix H.1.5 for a batch of 512 prompts. The measurements are averaged over 100 batches. Methods are ascendingly ordered by winrate. To the right of the dashed line, PPO, RLOO ($k = 4$), and REBEL have the highest winrates, which are comparable among them.

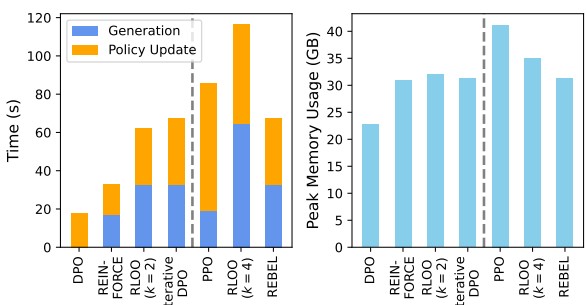

Figure 2: **Plot of runtime and memory usage.** Baselines on the left-hand side of the dashed line have lower winrates. Methods on the right-hand side of the dashed line have similar winrates to REBEL.

While DPO and REINFORCE are more time and memory-efficient, their performance does not match up to REBEL, as shown in Table 1. RLOO ($k = 2$) and Iterative DPO have similar runtime and memory usage as REBEL since we set $\mu = \pi_t$, making REBEL also generate twice per prompt. However, both methods have worse performance than REBEL. Compared to PPO and RLOO ($k = 4$), REBEL demonstrates shorter runtimes and lower peak memory usage. PPO is slow and requires more memory since it needs to update two networks (the policy network and the value network). RLOO ($k = 4$) requires four generations per prompt which makes it slow and less memory efficient. In summary, *compared to the two baselines (PPO and RLOO ($k = 4$)) that achieve similar winrates as REBEL, REBEL is more computationally tractable and simpler to implement.*

## 5.2 General Chat

**Task.** We consider a general chat scenario where $x$ is a prompt from the user and $y$ is a response. We adapt the setting from Zhu et al. (2023), using OpenChat-3.5 (Wang et al., 2024) as the base model, Starling-RM-7B-alpha (Zhu et al., 2023) as the reward model, and the Nectar dataset (Zhu et al., 2023). This setup enables a direct comparison between REBEL and APA (Zhu et al., 2023) which is used to train Starling-LM-7B-alpha.

**Evaluation.** Following previous works, we use AlpacaEval 2.0 (Dubois et al., 2024), MT-Bench (Zheng et al., 2023), and Open LLM Leaderboard (Beeching et al., 2023) as metrics. AlpacaEval 2.0 uses prompts from AlpacaFarm (Dubois et al., 2024) to compare model responses against a reference response generated by GPT-4-Turbo. We report the winrate over the reference responses. MT-Bench consists of 80 open-ended questions on various topics. Answers are scored directly by GPT-4. Open LLM Leaderboard consists of MMLU (Hendrycks et al., 2021), GSM8K (Cobbe et al., 2021), Arc (Clark et al., 2018), Winogrande (Sakaguchi et al., 2019), TruthfulQA (Lin et al., 2022), and HellaSwag (Zellers et al., 2019). The prompts of the tasks consist of zero or few-shot samples.

**Quality Analysis.** The results between models trained with REBEL and baseline methods are shown in Table 2. For MT-Bench and AlpacaEval 2.0, under the same setup, REBEL outperforms APA (Zhu et al., 2023) on both metrics, demonstrating the effectiveness of REBEL under chat setting and *its superior performance over APA*. For the metrics on Open LLM Leaderboard, REBEL is able to enhance the performance of GSM8K and HellaSwag and maintain the overall average as the base models. Similar values on MMLU as base models indicate that we preserve the basic capability of the pre-trained model during the RL fine-tuning process. We include a breakdown of MT-Bench in Appendix M.

### 5.2.1 Ablation: batch size and data sampling distributions

**Task.** In the previous section, we sample $y$ and $y'$ from $\pi_t(\cdot|x)$ and we use small batch size with $|\mathcal{D}_t| = 32$. In this section, we investigate the alternative sampling distribution described in Remark 1. Specifically, at each iteration, we generate 5 responses from $\pi_t$ for each prompt in the *entire dataset* (i.e., $|\mathcal{D}_t|$ is the size of the entire dataset), rank them based on the reward model, and set $y$ to be the best of the five responses, and $y'$ to be the worst of the five responses. We perform 3 iterations for this setup with Meta-Llama-3-8B-Instruct (Meta, 2024) as the base model, ArmoRM-Llama3-

| Method | MT-Bench | AlpacaEval 2.0 | | MMLU | GSM8K | Arc | Winogrande | TruthfulQA | HellaSwag |
| | | LC Win Rate | Win Rate | (5-shot) | (5-shot) | (25-shot) | (5-shot) | (0-shot) | (10-shot) |
|---|---|---|---|---|---|---|---|---|---|
| Base | 7.69 | 12.2 | 11.7 | 63.6 | 68.5 | **64.9** | 80.6 | 47.3 | 84.7 |
| APA | 7.43 | 14.7 | **14.2** | 63.4 | 68.0 | **64.9** | **81.1** | 47.3 | 84.8 |
| REBEL | **8.06** | **17.3** | 12.8 | **63.7** | **68.8** | 64.3 | 80.4 | **48.2** | **85.0** |

Table 2: **Results on General Chat.** The best-performing method for each metric is highlighted in bold. Note that the APA result is directly obtained by evaluating the Starling-LM-7B-alpha model.

| Method | MT-Bench | AlpacaEval 2.0 | | MMLU | GSM8K | Arc | Winogrande | TruthfulQA | HellaSwag | AH |
| | | LC Win Rate | Win Rate | (5-shot) | (5-shot) | (25-shot) | (5-shot) | (0-shot) | (10-shot) | |
|---|---|---|---|---|---|---|---|---|---|---|
| Base | 8.10 | 22.9 | 22.6 | 65.8 | 75.3 | **62.0** | 75.5 | 51.7 | 78.7 | 22.3 |
| DPO | 8.11 | 44.9 | 41.6 | 66.1 | 74.6 | 61.3 | 75.5 | **51.8** | **78.9** | 34.0 |
| REBEL (iter 1) | **8.13** | 48.3 | 41.8 | **66.3** | 75.8 | 61.7 | **75.9** | 51.8 | 78.7 | **34.5** |
| REBEL (iter 2) | 8.07 | **50.0** | **48.5** | 65.9 | 75.4 | 61.3 | 75.5 | 50.3 | 78.6 | 30.4 |
| REBEL (iter 3) | 8.01 | 49.7 | 48.1 | 66.0 | 75.7 | 61.1 | 75.7 | 49.8 | 78.8 | 30.0 |

Table 3: **Ablation Results.** In this table, REBEL uses a *larger batch size (the entire dataset) with the best-of-N and worst-of-N ($N = 5$) of $\pi_t$ as the sampling distributions for generating pairs $y, y'$*. The best-performing method for each metric is highlighted in bold. Note that DPO is trained on the *online data* generated by the base model and labeled by the RM.

8B-v0.1 (Wang et al.) as the reward model, and the UltraFeedback dataset (Cui et al., 2023). We compare REBEL with DPO which is also trained for one epoch on the entire dataset with best-of-5 as $y_w$ and worst-of-5 as $y_l$ sampled from $\pi_0$. In other words, the training data used for the first iteration of REBEL is the same as the one we use for DPO[3]. We follow the same evaluation methods as the previous section and include Arena Hard (AH) (Li et al., 2024) in our analysis.

**Quality Analysis.** Results in Table 3 show that REBEL can significantly improve the base model's performance, especially on AlpacaEval 2.0 and Arena Hard. Compared to DPO, the model trained by REBEL with 1 iteration is better in almost all datasets, demonstrating the benefit of using the fine-grained reward gap in policy optimization over just the zero-one labels. In this large batch setting, we find that more iterations in general do not help performance. We conjecture that this is the issue of overfitting to the training dataset. A more diverse and larger dataset can potentially address this issue.

### 5.3 Image Generation

**Task.** We also consider the setting of image generation, where, given a consistency model (Song et al., 2023) and a target reward function, we seek to train the consistency model to output images that garner a higher reward. We use 45 common animals as generation prompts similar to Black et al. (2023); Oertell et al. (2024) and the latent consistency model (Luo et al., 2023) distillation of the Dreamshaper v7 model, a finetune of stable diffusion (Rombach et al., 2021). We compare REBEL to a clipped, policy gradient objective (Black et al., 2023; Fan et al., 2024; Oertell et al., 2024) with the aim to optimize aesthetic quality to obtain a high reward from the LAION aesthetic score predictor (Schuhmann, 2022). This baseline does not use critics or GAE for advantage estimates. However, the clipping objective is clearly motivated by PPO, and thus, we simply name this baseline as PPO.

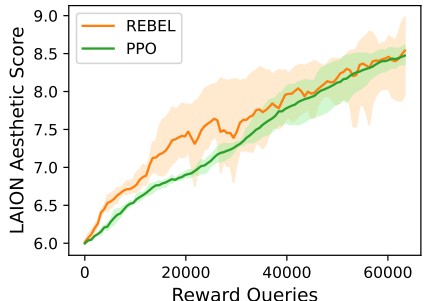

**Evaluation.** We evaluate on the reward under the LAION aesthetic reward model for an equal number of reward queries/samples generated and an equal number of gradient updates. The aesthetic predictor is trained to predict human-labeled scores of images on a scale of 1 to 10. Images that tend to have the highest reward are artwork. Following Agarwal et al. (2021), we report inter-quartile means (IQM) with 95% confidence intervals (CIs) across three seeds for both REBEL and PPO. The CIs were calculated with percentile bootstrap with stratified sampling over three random seeds.

Figure 3: **Learning curves** as a function of reward queries to the LAION aesthetic predictor. The colored areas represent 95% CIs.

---

[3]Directly training DPO on the original Ultrafeedback preference dataset does not provide strong performance under AlpacaEval (e.g., the LC-winrate is around 28%, see Song et al. (2024)). So for a fair comparison to REBEL, we train DPO on the online data generated by $\pi_0$ and labeled by the reward model.

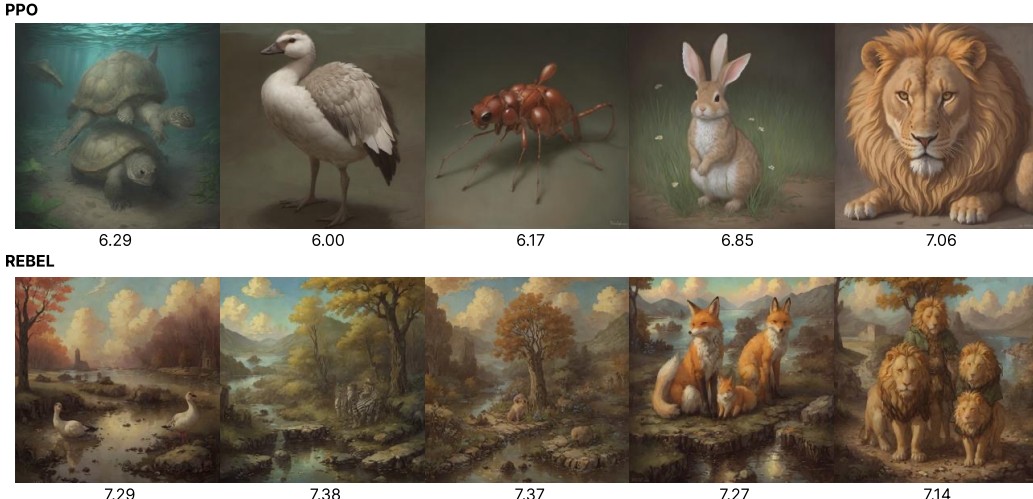

Figure 4: **Generated images** using PPO and REBEL during an intermediate checkpoint. At the same number of epochs, REBEL observes a higher reward under the reward model. This can further be seen by the more diverse background of images generated from REBEL with less training time.

**Quality Analysis.** Figure 3 shows REBEL optimizes the consistency model faster during the beginning of training and eventually achieves a performance similar to that of PPO. For our experiments, we tuned both batch size and learning rate for our algorithms, testing batch sizes of $[4, 8, 16]$ per GPU and learning rates $[1e-4, 3e-4, 6e-4, 1e-3]$. The main difference in implementation between PPO and REBEL is the replacement of the clipped PPO objective with our regression objective. To maximize LAION-predicted aesthetic quality, both REBEL and PPO transform a model that produces plain images into one that produces artistic drawings. We found across multiple seeds that REBEL produced lush backgrounds when compared to PPO's generations. Please see Appendix I.3 for more examples of generated images.

## 6 Conclusion and Future Work

We propose REBEL, an RL algorithm that reduces the problem of RL to solving a sequence of relative reward regression problems on iteratively collected datasets. In contrast to policy gradient approaches that require additional networks and heuristics like clipping to ensure optimization stability, it suffices for REBEL to merely drive down error on a least squares problem, making it strikingly simple to implement and scale. In theory, REBEL matches the best guarantees we have for RL algorithms in the agnostic setting, while in practice, REBEL is able to match and sometimes outperform methods that are far more complex to implement or expensive to run across both language modeling and guided image generation tasks.

There are several open questions raised by our work. The first is whether using a loss function other than square loss (e.g. log loss or cross-entropy) could lead to better performance in practice (Farebrother et al., 2024) or tighter bounds (e.g. first-order / gap-dependent) in theory (Foster and Krishnamurthy, 2021; Wang et al., 2023, 2024). The second is whether, in the general (i.e. non-utility-based) preference setting, the coverage condition assumed in our analysis is necessary – we conjecture it is. Relatedly, it would be interesting to explore whether using *preference* (rather than reward) models to provide supervision for REBEL replicates the performance improvements reported by Swamy et al. (2024); Munos et al. (2023); Calandriello et al. (2024). Third, while we focus primarily on the bandit setting in the preceding sections, it would be interesting to consider the more general RL setting and explore how offline datasets can be used to improve the efficiency of policy optimization via techniques like resets (Bagnell et al., 2003; Ross and Bagnell, 2014; Swamy et al., 2023; Chang et al., 2023, 2024; Ren et al., 2024; Dice et al., 2024).

## Acknowledgements

ZG and JDC are supported by LinkedIn under the LinkedIn-Cornell Grant. GKS is supported by his family and friends. KB is supported by NSF under grant No. 2127309 to the Computing Research Association for the CIFellows Project. JDL acknowledges support of the NSF CCF 2002272, NSF IIS 2107304, and NSF CAREER Award 2144994. WS acknowledges funding from NSF IIS-2154711, NSF CAREER 2339395, DARPA LANCER: LeArning Network CybERagents, and Cornell Infosys Collaboration.

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

# A   Detailed Discussion of Related Work

**Policy Gradients.** Policy gradient (PG) methods (Nemirovskij and Yudin, 1983; Williams, 1992; Konda and Tsitsiklis, 1999; Kakade, 2001; Schulman et al., 2017) are a prominent class of RL algorithms due to their direct, gradient-based policy optimization, robustness to model misspecification (Agarwal et al., 2020), and scalability to modern AI applications from fine-tuning LLMs (Stiennon et al., 2022) to optimizing text-to-image generators (Oertell et al., 2024).

Broadly speaking, we can taxonomize PG methods into two families. The first family is based on REINFORCE (Williams, 1992) and often includes variance reduction techniques (Kool et al., 2019; Richter et al., 2020; Zhu et al., 2023). While prior work by Ahmadian et al. (2024) has shown that REINFORCE-based approaches can outperform more complex RL algorithms like PPO on LLM fine-tuning tasks like *TL;DR*, we find that a properly optimized version of PPO still out-performs a REINFORCE baseline. The second family is *adaptive* PG techniques that *precondition* the policy gradient (usually with the inverse of the Fisher Information Matrix) to ensure it is *covariant* to re-parameterizations of the policy, which include NPG (Kakade, 2001; Bagnell and Schneider, 2003) and its practical approximations like TRPO (Schulman et al., 2015) and PPO (Schulman et al., 2017). Intuitively, the preconditioning ensures that we make small changes in terms of action distributions, rather than in terms of the actual policy parameters, leading to faster and more stable convergence. Unfortunately, computing and then inverting the Fisher Information Matrix is computationally intensive and therefore we often resort to approximations in practice, as done in TRPO. However, these approximations are still difficult to apply to large-scale generative models, necessitating even coarser approximations like PPO. In contrast, REBEL does not need any such approximations to be implemented at scale, giving us a much closer connection between theory and practice.

**Reward Regression.** The heart of REBEL is a novel reduction from RL to iterative squared loss regression. While using regression to fit either the reward (Peters and Schaal, 2007) or the value (Peng et al., 2019) targets which are then used to extract a policy have previously been explored, our method instead takes a page from DPO (Rafailov et al., 2023; Zhou et al., 2023) and inverse RL methods (Jacq et al., 2019; Watson et al., 2023) to implicitly parameterize the reward regressor in terms of the policy. This collapses the two-stage procedure of prior methods into a single step.

**Preference Fine-Tuning (PFT) of Generative Models.** RL has attracted renewed interest due to its central role in "aligning" language models – i.e., adapting their distribution of prompt completions towards the set of responses preferred by human raters.

One family of techniques for PFT, often referred to as Reinforcement Learning from Human Feedback (RLHF) involves first fitting a reward model (i.e. a classifier) to the human preference data and then using this model to provide reward values to a downstream RL algorithm (often PPO) (Christiano et al., 2017; Ziegler et al., 2020). LLMs fine-tuned by this procedure include GPT-N (OpenAI, 2023), Claude-N (Anthropic, 2024), and Llama-N (Meta, 2024). Similar approaches have proved beneficial for tasks like summarization (Stiennon et al., 2022), question answering (Nakano et al., 2022), text-to-image generation (Lee et al., 2023), and instruction following (Ouyang et al., 2022).

Another family of techniques for PFT essentially treats the problem as supervised learning and uses a variety of ranking loss functions. It includes DPO (Rafailov et al., 2023), IPO (Azar et al., 2023), and KTO (Ethayarajh et al., 2023). These techniques are simpler to implement as they remove components like an explicit reward model, value network, and on-policy training from the standard RLHF setup. However, recent work finds their performance to be lesser than that of on-policy methods (Lambert et al., 2024; Tajwar et al., 2024), which agrees with our findings. This is perhaps caused by their lack of interaction during training, leading to the well-known covariate shift/compounding error issue (Ross et al., 2011; Swamy et al., 2021) and the associated lower levels of performance.

The third family of PFT techniques combines elements from the previous two: it involves running an offline algorithm *iteratively*, collecting on-policy preference feedback from either a supervisor model such as GPT4 (Rosset et al., 2024; Xiong et al., 2024; Guo et al., 2024) or from a preference model fit on human data (Calandriello et al., 2024). All of these approaches can be considered instantiations of the general SPO reduction proposed by Swamy et al. (2024), which itself can be thought of as a preference-based variant of DAgger (Ross et al., 2011). Recent work by Tajwar et al. (2024) confirms the empirical strength of these techniques which leverage additional online data. Our approach fits best into this family of techniques – we also iteratively update our model by solving a sequence of supervised learning problems over on-policy datasets. However, REBEL comes with several key

differentiating factors from the prior work. Online versions of DPO or IPO (Xiong et al., 2024; Tajwar et al., 2024; Guo et al., 2024; Calandriello et al., 2024; Munos et al., 2023) essentially use a reward / preference model to generate binary win-loss labels while REBEL actually uses the output of the reward model as a regression target, taking advantage of this more nuanced feedback. In contrast to Rosset et al. (2024), algorithmically, REBEL does not use any online preference feedback from GPT4 nor does it require to generate a large number of responses per prompt, both of which are extremely expensive as reported by Rosset et al. (2024). Theoretically, we are able to prove policy performance bounds under a much weaker coverage condition. Unlike Mao et al. (2024) that regularize to the initial policy $\pi_0$ during updates, we perform *conservative* updates by regularizing $\pi_{t+1}$ to $\pi_t$. When doing the former, it is difficult to prove convergence or monotonic improvement as the current policy can just bounce around a ball centered at $\pi_0$, a well-known issue in the theory of approximate policy iteration (Kakade and Langford, 2002; Munos, 2003). In contrast, by incorporating the prior policy's probabilities into our regression problem, we are able to prove stronger guarantees for REBEL. When applying REBEL to the general preference setting, our algorithm shares some similarities with the concurrent work of Wu et al. (2024), which also leverages the DPO reparameterization trick to cleanly implement an Online Mirror Descent approach to computing a minimax winner via self-play. The key difference is that REBEL uses paired responses, while the algorithm of Wu et al. (2024) does not. Using paired responses, REBEL is able to cancel out the partition function and therefore does not need to resort to heuristic approximations of it (specifically, assuming the partition function is always equal to a constant). Furthermore, we can run REBEL with datasets consisting of a mixture of on-policy and off-policy data with strong guarantees, enabling *hybrid training*, as previously explored in the RL (Song et al., 2023; Ball et al., 2023; Zhou et al., 2023) and inverse RL (Ren et al., 2024) literature.

# B  Proof of Claim 1

We prove claim 1 in this section. We start from deriving the Fisher information matrix.

$$F_t := \frac{1}{\eta^2} \mathbb{E}_{x,y\sim\pi_t,y'\sim\mu} \left(\nabla_\theta \ln \pi_{\theta_t}(y|x) - \nabla_\theta \ln \pi_{\theta_t}(y'|x)\right) \left(\nabla_\theta \ln \pi_{\theta_t}(y|x) - \nabla_\theta \ln \pi_{\theta_t}(y'|x)\right)^\top$$

$$= \frac{2}{\eta^2} \mathbb{E}_{x,y\sim\pi_{mix}} \nabla_\theta \ln \pi_{\theta_t}(y|x) \nabla_\theta \ln \pi_{\theta_t}(y|x)^\top$$

where the last equality uses the fact that cross terms from completing the square are zero. Now recall Eq. 9 which is an ordinarily least square regression problem. The minimum norm solution of the least square regression problem is:

$$\delta = (\eta/2)\tilde{F}_t^\dagger \left(\mathbb{E}_{x,y\sim\pi_t,y'\sim\mu}\left(\nabla_\theta \ln \pi_{\theta_t}(y|x) - \nabla_\theta \ln \pi_{\theta_t}(y'|x)\right)(r(x,y) - r(x,y'))\right)$$

$$= (\eta/2)\tilde{F}_t^\dagger \Big(\mathbb{E}_{x,y\sim\pi_t}\left[\nabla_\theta \ln \pi_{\theta_t}(y|x)r(x,y)\right] + \mathbb{E}_{x,y'\sim\mu}\left[\nabla_\theta \ln \pi_{\theta_t}(y'|x)r(x,y')\right]$$

$$- \mathbb{E}_{x,y\sim\pi_t,y'\sim\mu}\nabla_\theta \ln \pi_{\theta_t}(y'|x)r(x,y)\Big)$$

$$= (\eta/2)\tilde{F}_t^\dagger \Big(\mathbb{E}_{x,y\sim\pi_t}\left[\nabla_\theta \ln \pi_{\theta_t}(y|x)[r(x,y) - \mathbb{E}_{y'\sim\pi_t(\cdot|x)}r(x,y')]\right]$$

$$+ \mathbb{E}_{x,y\sim\mu}\left[\nabla_\theta \ln \pi_{\theta_t}(y|x)[r(x,y) - \mathbb{E}_{y'\sim\pi_t(\cdot|x)}r(x,y')]\right]\Big)$$

$$= (\eta)\tilde{F}_t^\dagger \left(\mathbb{E}_{x,y\sim(\pi_t+\mu)/2}\left[\nabla_\theta \ln \pi_{\theta_t}(y|x)[A^{\pi_t}(x,y)]\right]\right)$$

where we again use the fact that $\mathbb{E}_{y\sim\pi_{\theta_t}(\cdot|x)}\nabla_\theta \ln \pi_{\theta_t}(y|x)g(x) = 0$ for any function $g(x)$, and we define *Advantage* $A^\pi(x,y) := r(x,y) - \mathbb{E}_{y'\sim\pi(\cdot|x)}r(x,y')$.

## C  Proof of Claim 2

We prove claim 2 in this section. We start by approximating our predictor $\frac{1}{\eta}\ln \pi_\theta(y|x)/\pi_{\theta_t}(y|x)$ by its first order Taylor expansion at $\theta_t$: $\frac{1}{\eta}\left(\ln \pi_\theta(y|x) - \ln \pi_{\theta_t}(y|x)\right) \approx \frac{1}{\eta}\nabla_\theta \ln \pi_{\theta_t}(y|x)^\top(\theta - \theta_t)$, where $\approx$ indicates that we ignore higher order terms in the expansion. Setting $\delta := \theta - \theta_t$ and replace $\frac{1}{\eta}\left(\ln \pi_\theta(y|x) - \ln \pi_{\theta_t}(y|x)\right)$ by its first order approximation in Eq. 1, we arrive at:

$$\min_\delta \sum_{(x,y,y')\in\mathcal{D}_t} \left(\frac{1}{\eta}\left(\nabla_\theta \ln \pi_{\theta_t}(y|x) - \nabla_\theta \ln \pi_{\theta_t}(y'|x)\right)^\top \delta - (r(x,y) - r(x,y'))\right)^2 \quad (11)$$

under finite setting.

Following the previous derivation, we have the unbiased estimate of Fisher information matrix under the finite setting as:

$$\hat{F}_t := \frac{2}{\eta^2 N}\sum_{x_n,y_n\sim\pi_{mix}} \nabla_\theta \ln \pi_{\theta_t}(y_n|x_n)\nabla_\theta \ln \pi_{\theta_t}(y_n|x_n)^\top$$

Since Eq. 11 is an ordinarily least square regression problem. The minimum norm solution of the least square regression problem is:

$$\delta = (\eta/2)\tilde{\hat{F}}_t^\dagger \frac{1}{N}\sum_n \left(\nabla_\theta \ln \pi_{\theta_t}(y_n|x_n) - \nabla_\theta \ln \pi_{\theta_t}(y'_n|x_n)\right)\left(r(x_n,y_n) - r(x_n,y'_n)\right)$$

$$= \eta\tilde{\hat{F}}_t^\dagger \frac{1}{2N}\sum_n \left(\nabla \ln \pi_{\theta_t}(y_n|x_n)(r(x_n,y_n) - r(x_n,y'_n)) + \nabla \ln \pi_{\theta_t}(y'_n|x_n)(r(x_n,y'_n) - r(x_n,y_n))\right)$$

# D  Extending `REBEL` to General Preferences

In the above discussion, we assume we are given access to a ground-truth reward function. However, in the generative model fine-tuning applications of RL, we often need to learn from human *preferences*, rather than rewards. This shift introduces a complication: not all preferences can be rationalized by an underlying utility function. In particular, *intransitive* preferences which are well-known to result from aggregation of different sub-populations or users evaluating different pairs of items on the basis of different features (May, 1954; Tversky, 1969; Gardner, 1970) cannot be accurately captured by a single reward model. To see this, note that if we have $a \succ b$, $b \succ c$, and $c \succ a$, it is impossible to have a reward model that simultaneously sets $\hat{r}(a) > \hat{r}(b)$, $\hat{r}(b) > \hat{r}(c)$, and $\hat{r}(c) > \hat{r}(a)$. As we increase the space of possible choices to that of all possible prompt completions, the probability of such intransitivities sharply increases (Dudík et al., 2015), as reflected in the high levels of annotator disagreement in LLM fine-tuning datasets (Touvron et al., 2023). Thus, rather than assuming access to a reward model, in such settings, we assume access to a *preference model* (Munos et al., 2023; Swamy et al., 2024; Rosset et al., 2024; Ye et al., 2024).

## D.1  A Game-Theoretic Perspective on Learning from Preferences

More specifically, for any tuple $(x, y, y')$, we assume we have access to $\mathcal{P}(y \succ y'|x)$: the probability that $y$ is preferred to $y'$. We then define our preference model $l$ as

$$l(x, y, y') \triangleq 2 \cdot \mathcal{P}(y \succ y'|x) - 1. \tag{12}$$

Observe that $l(x, y, y') \in [-1, 1]$ is skew-symmetric, i.e., $l(x, y, y) = 0$, $l(x, y, y') + l(x, y', y) = 0$ for all $x \in \mathcal{X}, y, y' \in \mathcal{Y}$. If the learner can only receive a binary feedback $o \in \{0, 1\}$ indicating the preference between $y$ and $y'$, we assume $o$ is sampled from a Bernoulli distribution with mean $\mathcal{P}(y \succ y'|x)$, where $o = 1$ means that $y$ is preferred over $y'$ and $0$ otherwise.

Given access to such a preference model, a solution concept to the preference aggregation problem with deep roots in the social choice theory literature (Kreweras, 1965; Fishburn, 1984; Kramer, 1973; Simpson, 1969) and the dueling bandit literature (Yue et al., 2012; Dudík et al., 2015) is that of a minimax winner (MW) $\pi_{\mathsf{MW}}$: the Nash Equilibrium strategy of the symmetric two-player zero-sum game with $l$ as a payoff function. In particular, due to the skew-symmetric property of $l$, Swamy et al. (2024) proved that there exists a policy $\pi_{\mathsf{MW}}$ such that

$$\max_{\pi} \mathbb{E}_{x \sim \rho, y \sim \pi(\cdot|x), y' \sim \pi_{\mathsf{MW}}(\cdot|x)} \left[ l(x, y, y') \right] = \min_{\pi} \mathbb{E}_{x \sim \rho, y \sim \pi_{\mathsf{MW}}(\cdot|x), y' \sim \pi(\cdot|x)} \left[ l(x, y, y') \right].$$

This implies that $(\pi_{\mathsf{MW}}, \pi_{\mathsf{MW}})$ is a Nash Equilibrium (Wang et al., 2023; Munos et al., 2023; Swamy et al., 2024; Ye et al., 2024). As is standard in game solving, our objective is to obtain an $\epsilon$-approximate MW $\hat{\pi}$ measured by the duality gap (DG):

$$\mathsf{DG}(\hat{\pi}) := \max_{\pi} \mathbb{E}_{x \sim \rho, y \sim \pi(\cdot|x), y' \sim \hat{\pi}(\cdot|x)} \left[ l(x, y, y') \right] - \min_{\pi} \mathbb{E}_{x \sim \rho, y \sim \hat{\pi}(\cdot|x), y' \sim \pi(\cdot|x)} \left[ l(x, y, y') \right] \leq \epsilon.$$

In the following discussion, we will use $l(x, y, \pi)$ to denote $\mathbb{E}_{y' \sim \pi(\cdot|x)}[l(x, y, y')]$ and $l(\pi, \pi')$ to denote $\mathbb{E}_{x \sim \rho, y \sim \pi(\cdot|x), y' \sim \pi'(\cdot|x)}[l(x, y, y')]$ for notational convenience.

## D.2  Self-Play Preference Optimization (SPO) with `REBEL` as Base Learner

We can straightforwardly extend `REBEL` to the general preference setting via an instantiation of the Self-Play Preference Optimization (SPO) reduction of Swamy et al. (2024). In short, Swamy et al. (2024) prove that rather than performing adversarial training, we are able to perform a simple and stable *self-play* procedure while retaining strong theoretical guarantees. Practically, this corresponds to sampling at leas two completions from the current policy, querying a learned preference / supervisor model on each pair, and using the winrate for each completion as its reward. We will now describe how we can adapt `REBEL` to this mode of feedback.

Assuming that we can query the preference oracle $l(x, y, y')$ at will, we can modify the least square objective Eq. (1) to

$$\theta_{t+1} := \operatorname*{argmin}_{\theta} \sum_{x, y, y', y'' \in \mathcal{D}_t} \left( \frac{1}{\eta} \left( \ln \frac{\pi_{\theta}(y|x)}{\pi_{\theta_t}(y|x)} - \ln \frac{\pi_{\theta}(y'|x)}{\pi_{\theta_t}(y'|x)} \right) - \left( l(x, y, y'') - l(x, y', y'') \right) \right)^2$$

where $x \sim \rho, y \sim \pi_t(\cdot|x), y'' \sim \pi_t(\cdot|x), y' \sim \mu(\cdot|x)$. When the exact value of $l(x, y, y')$ is unavailable but only a binary preference feedback $o_{y,y'} \in \{0, 1\}$ sampling from Bernoulli with mean $l(x, y, y')$ is available, we can just replace $l(x, y, y'') - l(x, y', y'')$ by $o_{y,y'} - o_{y',y''}$. It is easy to see that the Bayes optimal of the above least square regression problem is equal to:

$$\mathbb{E}_{y'' \sim \pi_t(\cdot|x)} l(x, y, y'') - \mathbb{E}_{y'' \sim \pi_t(\cdot|x)} l(x, y', y'') = l(x, y, \pi_t) - l(x, y', \pi_t).$$

Swamy et al. (2024) define an iteration-dependent reward $r_t(x, y) := \mathbb{E}_{y'' \sim \pi_t(\cdot|x)} l(x, y, y'') = l(x, y, \pi_t)$. Thus, the above regression problem can be understood as an extension of REBEL to the setting where the reward function changes at each iteration $t$. Swamy et al. (2024) shows that running the exact MD (Eq. 3) with this iteration-dependent reward function $r_t$ leads to fast convergence to an approximate Minimax Winner, a property that we will use to provide the regret bound of REBEL in the general preference setting while accounting for nonzero mean squared error.

# E   Justification for Assumption 1

Intuitively, this assumption is saying that there is a function in our class of regressors that is able to accurately fit the difference of rewards. Recall that our class of regressors is isomorphic to our policy class. Therefore, as long as our class of policies is expressive, we would expect this assumption to hold with small $\epsilon$. For all domains we consider, our policy class is a flexible set of generative models (e.g. Transformer-based LLMs or diffusion models). Thus, we believe it is reasonable to believe this assumption holds in practice – see Figure 6 in Appendix L for empirical evidence of this point and Example 1 for more discussion.

More formally, the above assumption bounds the standard **in-distribution generalization error** (v.s. the point-wise guarantee in Eq. 8) of a well-defined supervised learning problem: least squares regression. The generalization error $\epsilon$ captures the possible errors from the learning process for $\theta_{t+1}$ and it could depend on the complexity of the policy class and the number of samples used in the dataset $\mathcal{D}_t$. For instance, when the the function $\ln \pi - \ln \pi'$ induced by the log-difference of two policies $(\pi, \pi')$ are rich enough (e.g., policies are deep neural networks) to capture the reward difference, then $\epsilon$ in this assumption converges to zero as we increase the number of training data. Note that while $\epsilon$ can be small, it does *not* imply that the learned predictor will have a small prediction error in a point-wise manner – it almost certainly will not.

**Example 1.** *One simple example is when $\pi(y|x) \propto \exp(\theta^\top \phi(x,y))$ for some features $\phi(x,y)$. In this case, $\ln(\pi(y|x)/\pi_t(y|x)) - \ln(\pi(y'|x)/\pi_t(y'|x)) = (\theta - \theta_t)^\top (\phi(x,y) - \phi(x,y'))$, which means that our regression problem in Eq. 1 is a classic linear regression problem. When the reward $r(x,y)$ is also linear in feature $\phi(x,y)$, then Eq. 1 is a well-specified linear regression problem, and $\epsilon$ typically scales in the rate of $O\left(d/|\mathcal{D}_t|\right)$ with $d$ being the dimension of feature $\phi$.*

*We can extend the above example to the case where $\phi$ is the feature corresponding to some kernel, e.g., RBF kernel or even Neural Tangent Kernel, which allows us to capture the case where $\pi$ is a softmax wide neural network with the least square regression problem solved by gradient flow. The error $\epsilon$ again scales $poly(d/|\mathcal{D}_t|)$, where $d$ is the effective dimension of the corresponding kernel.*

# F Proof of Theorem 1

In this section, we provide the proof of theorem 1. For notation simplicity, throughout the proof, we denote $\pi_t$ for $\pi_{\theta_t}$, and define $f_t(x, y) := \frac{1}{\eta} \ln \frac{\pi_{t+1}(y|x)}{\pi_t(y|x)}$.

The following lemma shows that the learned function $f_t$ can predict reward $r$ well under both $\pi_t$ and $\mu$ up to terms that are $y$-independent.

**Lemma 1.** *Consider any $t \in [T]$. Define $\Delta(x, y) = f_t(x, y) - r(x, y)$. Define $\Delta_{\pi_t}(x) = \mathbb{E}_{y \sim \pi_t(\cdot|x)} \Delta(x, y)$ and $\Delta_\mu(x) = \mathbb{E}_{y \sim \mu(\cdot|x)} \Delta(x, y)$. Under assumption 1, for all t, we have the following:*

$$\mathbb{E}_{x, y \sim \pi_t(\cdot|x)} \left( f_t(x, y) - r(x, y) - \Delta_{\pi_t}(x) \right)^2 \leq \epsilon, \tag{13}$$

$$\mathbb{E}_{x, y \sim \mu(\cdot|x)} \left( f_t(x, y) - r(x, y) - \Delta_\mu(x) \right)^2 \leq \epsilon, \tag{14}$$

$$\mathbb{E}_x \left( \Delta_{\pi_t}(x) - \Delta_\mu(x) \right)^2 \leq \epsilon. \tag{15}$$

*Proof.* From assumption 1, we have:

$$\mathbb{E}_{x, y_1 \sim \pi_t, y_2 \sim \mu} \left( f_t(x, y_1) - \Delta_{\pi_t}(x) - r(x, y_1) - (f_t(x, y_2) - \Delta_\mu(x) - r(x, y_2)) + \Delta_{\pi_t}(x) - \Delta_\mu(x) \right)^2$$

$$= \mathbb{E}_{x, y_1 \sim \pi_t} \left( f_t(x, y_1) - \Delta_{\pi_t}(x) - r(x, y_1) \right)^2 + \mathbb{E}_{x, y_2 \sim \mu} \left( f_t(x, y_2) - \Delta_\mu(x) - r(x, y_2) \right)^2$$

$$- 2\mathbb{E}_{x, y_1 \sim \pi_t, y_2 \sim \mu} \left( f_t(x, y_1) - \Delta_{\pi_t}(x) - r(x, y_1) \right) \left( f_t(x, y_2) - \Delta_\mu(x) - r(x, y_2) \right)$$

$$+ 2\mathbb{E}_{x, y_1 \sim \pi_t} \left( f_t(x, y_1) - \Delta_{\pi_t}(x) - r(x, y_1) \right) \left( \Delta_{\pi_t}(x) - \Delta_\mu(x) \right)$$

$$- 2\mathbb{E}_{x, y_2 \sim \pi_t} \left( f_t(x, y_2) - \Delta_\mu(x) - r(x, y_2) \right) \left( \Delta_{\pi_t}(x) - \Delta_\mu(x) \right) + \mathbb{E}_x (\Delta_1(x) - \Delta_2(x))^2$$

$$= \mathbb{E}_{x, y_1 \sim \pi_t} \left( f_t(x, y_1) - \Delta_{\pi_t}(x) - r(x, y_1) \right)^2 + \mathbb{E}_{x, y_2 \sim \mu} \left( f_t(x, y_2) - \Delta_\mu(x) - r(x, y_2) \right)^2$$

$$+ \mathbb{E}_x (\Delta_{\pi_t}(x) - \Delta_\mu(x))^2 \leq \epsilon.$$

In the above, we first complete the square, and then we only keep terms that are not necessarily zero. Since all the remaining three terms are non-negative, this concludes the proof. □

By the definition of $f_t$, we have $\Delta(x, y) = \frac{1}{\eta} \ln \frac{\pi_{t+1}(y|x)}{\pi_t(y|x)} - r(x, y)$. Taking $\exp$ on both sides, we get:

$$\forall x, y : \pi_{t+1}(y|x) = \pi_t(y|x) \exp\left(\eta(r(x, y) + \Delta(x, y))\right) = \frac{\pi_t(y|x) \exp(\eta(r(x, y) + \Delta(x, y) - \Delta_\mu(x)))}{\exp(-\eta\Delta_\mu(x))}$$

Denote $g_t(x, y) := r(x, y) + \Delta(x, y) - \Delta_\mu(x)$, and the advantage $A_t(x, y) = g_t(x, y) - \mathbb{E}_{y' \sim \pi_t(\cdot|x)} g_t(x, y')$. We can rewrite the above update rule as:

$$\forall x, y : \pi_{t+1}(y|x) \propto \pi_t(y|x) \exp(\eta A_t(x, y)) \tag{16}$$

In other words, the algorithm can be understood as running MD on the sequence of $A_t$ for $t = 0$ to $T - 1$. The following lemma is the standard MD regret lemma.

**Lemma 2.** *Assume $\max_{x, y, t} |A_t(x, y)| \leq A \in \mathbb{R}^+$, and $\pi_0(\cdot|x)$ is uniform over $\mathcal{Y}$. Then with $\eta = \sqrt{\ln(|\mathcal{Y}|)/(A^2 T)}$, for the sequence of policies computed by REBEL, we have:*

$$\forall \pi, x : \sum_{t=0}^{T-1} \mathbb{E}_{y \sim \pi(\cdot|x)} A_t(x, y) \leq 2A\sqrt{\ln(|\mathcal{Y}|)T}.$$

*Proof.* For completeness, we provide the proof here. Start with $\pi_{t+1}(y|x) = \pi_t(y|x) \exp(\eta A_t(x, y))/Z_t(x)$ where $Z_t(x)$ is the normalization constant, taking log on both sides, and add $\mathbb{E}_{y \sim \pi(\cdot|x)}$, we have:

$$-\text{KL}(\pi(\cdot|x)||\pi_{t+1}(\cdot|x)) = -\text{KL}(\pi(\cdot|x)||\pi_t(\cdot|x)) + \eta\mathbb{E}_{y \sim \pi(\cdot|x)} A_t(x, y) - \mathbb{E}_{y \sim \pi(\cdot|x)} \ln Z_t(x).$$

Rearrange terms, we get:

$$-\text{KL}(\pi(\cdot|x)||\pi_t(\cdot|x)) + \text{KL}(\pi(\cdot|x)||\pi_{t+1}(\cdot|x) = \mathbb{E}_{y \sim \pi(\cdot|x)} \left[ -\eta A_t(x, y) + \ln Z_t(x) \right]$$

For $\ln Z_t(x)$, using the condition that $\eta \leq 1/A$, we have $\eta A_t(x, y) \leq 1$, which allows us to use the inequality $\exp(x) \leq 1 + x + x^2$ for any $x \leq 1$, which lead to the following inequality:

$$\ln Z_t(x) = \ln \left( \mathbb{E}_{y \sim \pi(\cdot|x)} \exp(\eta A_t(x, y)) \right)$$

$$\leq \ln \left( \sum_y \pi_t(y|x) \left( 1 + \eta A_t(x, y) + \eta^2 A_t(x, y)^2 \right) \right)$$

$$\leq \ln \left( 1 + 0 + \eta^2 A^2 \right) \leq \eta^2 A^2,$$

where the last inequality uses $\ln(1 + x) \leq x$, and we used the fact that $\mathbb{E}_{y \sim \pi_t(x)} A_t(x, y) = 0$ due to the definition of advantage $A_t$. Thus, we have:

$$-\mathrm{KL}(\pi(\cdot|x)||\pi_t(\cdot|x)) + \mathrm{KL}(\pi(\cdot|x)||\pi_{t+1}(\cdot|x)) \leq -\mathbb{E}_{y \sim \pi(\cdot|x)}[A_t(x, y)] + \eta^2 A^2.$$

Sum over all iterations and do the telescoping sum, we get:

$$\sum_{t=0}^{T-1} \mathbb{E}_{y \sim \pi(\cdot|x)} A_t(x, y) \leq \mathrm{KL}(\pi(\cdot|x)||\pi_0(\cdot|x))/\eta + T\eta A^2 \leq \ln(|\mathcal{Y}|)/\eta + T\eta A^2.$$

With $\eta = \sqrt{\ln(|\mathcal{Y}|)/(A^2 T)}$, we conclude the proof. $\qquad \square$

With the above, now we are ready to conclude the proof of the main theorem.

*Proof of Theorem 1.* Consider a comparator policy $\pi^*$. We start with the performance difference between $\pi^*$ and the uniform mixture policy $\bar{\pi} := \sum_{t=0}^{T-1} \pi_t/T$:

$$\frac{1}{T} \sum_{t=0}^{T-1} \left( \mathbb{E}_{x, y \sim \pi^*(\cdot|x)} r(x, y) - \mathbb{E}_{x, y \sim \pi_t(\cdot|x)} r(x, y) \right) = \frac{1}{T} \sum_{t=0}^{T-1} \mathbb{E}_{x, y \sim \pi^*(\cdot|x)} \left( A^{\pi_t}(x, y) \right),$$

where we define the real advantage $A^{\pi_t}(x, y) := r(x, y) - \mathbb{E}_{y \sim \pi_t(\cdot|x)} r(x, y)$. Continue, we have:

$$\frac{1}{T} \sum_{t=0}^{T-1} \mathbb{E}_{x, y \sim \pi^*(\cdot|x)} \left( A^{\pi_t}(x, y) \right)$$

$$= \frac{1}{T} \sum_{t=0}^{T-1} \mathbb{E}_{x, y \sim \pi^*(\cdot|x)} \left( A_t(x, y) \right) + \frac{1}{T} \sum_{t=0}^{T-1} \mathbb{E}_{x, y \sim \pi^*(\cdot|x)} \left( A^{\pi_t}(x, y) - A_t(x, y) \right)$$

$$\leq 2A \sqrt{\frac{\ln(|\mathcal{Y}|)}{T}} + \frac{1}{T} \sum_{t=0}^{T-1} \sqrt{\mathbb{E}_x \mathbb{E}_{y \sim \pi^*(\cdot|x)} (A^{\pi_t}(x, y) - A_t(x, y))^2}$$

where the last inequality uses Lemma 2. We now just need to bound $\mathbb{E}_{y \sim \pi^*(\cdot|x)} (A^{\pi_t}(x, y) - A_t(x, y))^2$.

$$\mathbb{E}_x \mathbb{E}_{y \sim \pi^*(\cdot|x)} (A^{\pi_t}(x, y) - A_t(x, y))^2 = \mathbb{E}_x \mathbb{E}_{y \sim \mu(\cdot|x)} \frac{\pi^*(y|x)}{\mu(y|x)} (A^{\pi_t}(x, y) - A_t(x, y))^2$$

$$\leq C_{\pi^*} \mathbb{E}_{x, y \sim \mu(\cdot|x)} (A^{\pi_t}(x, y) - A_t(x, y))^2$$

where the last inequality uses the definition of concentrability coefficient $C_{\pi^*}$. We now bound $\mathbb{E}_{x, y \sim \mu(\cdot|x)} (A^{\pi_t}(x, y) - A_t(x, y))^2$. Recall the definiton of $A_t$ from Lemma 2.

$$\mathbb{E}_{x, y \sim \mu(\cdot|x)} (A^{\pi_t}(x, y) - A_t(x, y))^2$$

$$= \mathbb{E}_{x, y \sim \mu(\cdot|x)} (r(x, y) - \mathbb{E}_{y' \sim \pi_t(\cdot|x)} r(x, y') - g_t(x, y) + \mathbb{E}_{y' \sim \pi_t(\cdot|x)} g_t(x, y'))^2$$

$$\leq 2 \mathbb{E}_{x, y \sim \mu(\cdot|x)} (r(x, y) - g_t(x, y))^2 + 2 \mathbb{E}_x \mathbb{E}_{y' \sim \pi_t(\cdot|x)} (r(x, y') - g_t(x, y'))^2$$

Recall the $g_t(x, y) = r(x, y) + \Delta(x, y) - \Delta_\mu(x)$, and from Lemma 1, we can see that

$$\mathbb{E}_{x, y \sim \mu(\cdot|x)} (r(x, y) - g_t(x, y))^2 = \mathbb{E}_{x, y \sim \mu(\cdot|x)} (\Delta(x, y) - \Delta_\mu(x))^2 \leq \epsilon.$$

For $\mathbb{E}_x \mathbb{E}_{y' \sim \pi_t(\cdot|x)} \left( r(x, y') - g_t(x, y') \right)^2$, we have:

$$\mathbb{E}_x \mathbb{E}_{y' \sim \pi_t(\cdot|x)} \left( r(x, y') - g_t(x, y') \right)^2 = \mathbb{E}_x \mathbb{E}_{y' \sim \pi_t(\cdot|x)} \left( \Delta(x, y') - \Delta_\mu(x) \right)^2$$

$$= \mathbb{E}_x \mathbb{E}_{y' \sim \pi_t(\cdot|x)} \left( \Delta(x, y') - \Delta_{\pi_t}(x) + \Delta_{\pi_t}(x) - \Delta_\mu(x) \right)^2$$

$$\leq 2\mathbb{E}_x \mathbb{E}_{y' \sim \pi_t(\cdot|x)} \left( \Delta(x, y') - \Delta_{\pi_t}(x) \right)^2 + 2\mathbb{E}_x \left( \Delta_{\pi_t}(x) - \Delta_\mu(x) \right)^2 \leq 4\epsilon,$$

where the last inequality uses Lemma 1 again. This step relies on the fact that one of the samples is always on-policy, i.e., from $\pi_t$.

Combine things together, we can conclude that:

$$\mathbb{E}_x \mathbb{E}_{y \sim \pi^*(\cdot|x)} (A^{\pi_t}(x, y) - A_t(x, y))^2 \leq C_{\pi^*}(10\epsilon).$$

Finally, for the regret, we can conclude:

$$\frac{1}{T} \sum_{t=0}^{T-1} \mathbb{E}_{x, y \sim \pi^*(\cdot|x)} (A^{\pi_t}(x, y)) \leq 2A\sqrt{\frac{\ln |\mathcal{Y}|}{T}} + \frac{1}{T} \sum_t \sqrt{C_{\pi^*} 10\epsilon} = 2A\sqrt{\frac{\ln |\mathcal{Y}|}{T}} + \sqrt{C_{\pi^*} 10\epsilon}.$$

$\square$

# G Extension of analysis to General Preferences

Extending the above analysis to the general preference case is straightforward except that it requires a stronger coverage condition. This is because we want to find a Nash Equilibrium, which requires a comparison between the learned policy against all the other policies. Results from the Markov Game literature (Cui and Du, 2022; Zhong et al., 2022; Cui and Du, 2022; Xiong et al., 2023) and Cui and Du (2022) have shown that the standard single policy coverage condition used in single-player optimization is provably not sufficient. In particular, they propose using a notion of *unilateral concentrability* for efficient learning, which can be defined as

$$C_{\mathsf{uni},\mu} := \max_{\pi,x,y,y''} \frac{\pi_{\mathsf{MW}}(y|x)\pi(y''|x)}{\mu(y|x)\mu(y''|x)},$$

in the general preference setting. Notably, the above unilateral concentrability coefficient $C_{\mathsf{uni},\mu}$ is equivalent to $C_\mu := \max_{\pi,x,y} \frac{\pi(y|x)}{\mu(y|x)}$ since $C_\mu \le C_{\mathsf{uni},\mu} \le C_\mu^2$. Therefore in the following discussion, we will use $C_\mu$ as the coverage condition. In addition, we also assume the generalization error of the regression problem is small,

**Assumption 2** (Regression generalization bounds for general preference). *Over $T$ iterations, assume that for all t, we have:*

$$\mathbb{E}_{x\sim\rho,y\sim\pi_t(\cdot|x),y'\sim\mu(\cdot|x)} \left( \frac{1}{\eta} \left( \ln\frac{\pi_{\theta_{t+1}}(y|x)}{\pi_{\theta_t}(y|x)} - \ln\frac{\pi_{\theta_{t+1}}(y'|x)}{\pi_{\theta_t}(y'|x)} \right) - (l(x,y,\pi_t) - l(x,y',\pi_t)) \right)^2 \le \epsilon,$$

*for some $\epsilon$.*

Under the above coverage condition and generalization bound, we can show that REBEL is able to learn an approximate Minimax Winner:

**Theorem 2.** *With assumption 2, after $T$ many iterations, with a proper learning rate $\eta$, the policy $\widehat{\pi} = \mathsf{Unif}(\{\pi_t\}_{t=1}^T)$ satisfies that:*

$$\mathsf{DG}(\widehat{\pi}) \le O\left( \sqrt{\frac{1}{T}} + \sqrt{C_\mu\epsilon} \right).$$

*Here the O-notation hides problem-dependent constants that are independent of $\epsilon, C_\mu, T$.*

Note that the coverage condition here is much stronger than the single policy coverage condition in the RL setting. We conjecture that this is the cost one has to pay by moving to the more general preference setting and leaving the investigation of the necessarily coverage condition for future work.

## G.1 Proof of Theorem 2

Recall that $r_t(x,y) = l(x,y,\pi_t)$. Let us define $\Delta^t(x,y) := f_t(x,y) - r_t(x,y)$, $\Delta^t_{\pi_t}(x) := \mathbb{E}_{y\sim\pi_t(\cdot|x)}\Delta^t(x,y)$ and $\Delta^t_\mu(x) := \mathbb{E}_{y\sim\mu(\cdot|x)}\Delta^t(x,y)$. Then following the same arguments in Lemma 1, we have

$$\mathbb{E}_{x\sim\rho,y\sim\pi_t(\cdot|x)} \left[ \left(f_t(x,y) - r_t(x,y) - \Delta^t_{\pi_t}(x)\right)^2 \right] \le \epsilon, \tag{17}$$

$$\mathbb{E}_{x\sim\rho,y\sim\mu(\cdot|x)} \left[ \left(f_t(x,y) - r_t(x,y) - \Delta^t_\mu(x)\right)^2 \right] \le \epsilon, \tag{18}$$

$$\mathbb{E}_{x\sim\rho} \left[ \left(\Delta^t_{\pi_t}(x) - \Delta^t_\mu(x)\right)^2 \right] \le \epsilon. \tag{19}$$

With slight abuse of the notation, We also use $g_t$ and $A_t(x,y)$ to denote $r_t(x,y)+\Delta^t(x,y)-\Delta^t_\mu(x,y)$ and $g_t(x,y) - \mathbb{E}_{y'\sim\pi_t(\cdot|x)}g_t(x,y')$. Then following the same arguments in Lemma 2,

$$\forall\pi, x: \sum_{t=0}^{T-1} \mathbb{E}_{y\sim\pi(\cdot|x)}A_t(x,y) \le 2A\sqrt{\ln(|\mathcal{Y}|)T}. \tag{20}$$

Note that we have

$$\max_{\pi} l(\pi, \widehat{\pi}) = \max_{\pi} \frac{1}{T} \sum_{t=1}^{T} l(\pi, \pi_t)$$

$$= \max_{\pi} \frac{1}{T} \sum_{t=1}^{T} \mathbb{E}_{x\sim\rho, y\sim\pi(\cdot|x)}[r_t(x,y)] = \max_{\pi} \frac{1}{T} \sum_{t=1}^{T} \mathbb{E}_{x\sim\rho, y\sim\pi(\cdot|x)}[A^{t,\pi_t}(x,y)],$$

where $A^{t,\pi_t} := r_t(x,y) - \mathbb{E}_{y\sim\pi_t(\cdot|x)}[r_t(x,y)]$. The last step is due to the skew symmetry of $l$, i.e., $\mathbb{E}_{y\sim\pi_t(\cdot|x)}[r_t(x,y)] = l(x, \pi_t, \pi_t) = 0$. Then by following the same arguments in the proof of Theorem 1, with (17)(18)(19)(20), we have for any policy $\pi$,

$$\frac{1}{T} \sum_{t=0}^{T-1} \mathbb{E}_{x\sim\rho, y\sim\pi(\cdot|x)} \left(A^{t,\pi_t}(x,y)\right) \leq 2A\sqrt{\frac{\ln|\mathcal{Y}|}{T}} + \sqrt{10 C_{\mu\to\pi}\epsilon}.$$

This implies that

$$\max_{\pi} l(\pi, \widehat{\pi}) \leq \max_{\pi} \left(2A\sqrt{\frac{\ln|\mathcal{Y}|}{T}} + \sqrt{10 C_{\mu\to\pi}\epsilon}\right) \leq 2A\sqrt{\frac{\ln|\mathcal{Y}|}{T}} + \sqrt{10 C_{\mu}\epsilon}.$$

Note that due to the skew symmetry of $l$, we have

$$\min_{\pi} l(\widehat{\pi}, \pi) = \min_{\pi} \mathbb{E}_{x\sim\rho, y\sim\widehat{\pi}(\cdot|x), y'\sim\pi(\cdot|x)} [l(x,y,y')] = -\max_{\pi} \mathbb{E}_{x\sim\rho, y\sim\widehat{\pi}(\cdot|x), y'\sim\pi(\cdot|x)} [-l(x,y,y')]$$

$$= -\max_{\pi} \mathbb{E}_{x\sim\rho, y\sim\pi(\cdot|x), y'\sim\widehat{\pi}(\cdot|x)} [l(x,y,y')] = -\max_{\pi} l(\pi, \widehat{\pi}) \geq -2A\sqrt{\frac{\ln|\mathcal{Y}|}{T}} - \sqrt{10 C_{\mu}\epsilon}.$$

Therefore we have

$$\mathsf{DG}(\widehat{\pi}) \leq 4A\sqrt{\frac{\ln|\mathcal{Y}|}{T}} + 2\sqrt{10 C_{\mu}\epsilon}.$$

# H Additional Experiment Details

## H.1 Summarization

### H.1.1 Dataset Details

We present dataset details in Table 4. Dataset available at https://github.com/openai/summarize-from-feedback

Table 4: Dataset split, prompts, and maximum generation length for *TL;DR* summarization

| Dataset | Train/Val/Test | Prompt | Generation Length |
|---|---|---|---|
| Human Reference | 117K/6.45K/6.55K | "TL;DR:" | 53 |
| Preference | 92.9K/83.8K/- | "TL;DR:" | 53 |

### H.1.2 Model Details

For SFT models, we train a Pythia 1.4B (Biderman et al., 2023)[4] model for 1 epoch over the dataset with human references as labels, and use the existing fine-tuned 2.8B[5] and 6.9B[6] models. For reward models, we train a Pythia 1.4B parameter model for 1 epoch over the preference dataset and use the existing reward models with 2.8B[7] and 6.9B[8] parameters. For both REBEL and baseline methods using 1.4B and 2.8B parameters, we trained the policy and/or the critic using **low-rank adapters (LoRA)** (Hu et al., 2022) on top of our SFT and/or reward model respectively. For the 6.9B models, we perform **full-parameter** training. The 1.4B and 2.8B models are trained on 8 A6000 GPUs for one day and two days respectively. The 6.9B model is train on 8 H100 GPUs for two days.

### H.1.3 Baseline Implementation Details

For supervised fine-tuning (SFT), reward modeling training, PPO, and DPO, we follow the implementation at https://github.com/vwxyzjn/summarize_from_feedback_details (Huang et al., 2024). For iterative dpo, we implement as follows:

---
**Algorithm 2** Iterative DPO
---
1: **Input**: Reward $r$, policy class $\Pi = \{\pi_\theta\}$, parameter $\beta$
2: Initialize policy $\pi_{\theta_0}$.
3: **for** $t = 0$ to $T - 1$ **do**
4:    Collect dataset $\mathcal{D}_t = \{x, y, y'\}$ where $x \sim \rho, y \sim \pi_t(\cdot|x), y' \sim \pi_t(\cdot|x)$
5:    Solve square loss regression problem:

$$\theta_{t+1} = \operatorname*{argmin}_\theta \sum_{(x,y,y')\in\mathcal{D}_t} - \left[ \ln \sigma \left( (\beta \ln \frac{\pi_\theta(y|x)}{\pi_{\theta_t}(y|x)} - \beta \ln \frac{\pi_\theta(y'|x)}{\pi_{\theta_t}(y'|x)}) sgn\left(r(x,y) - r(x,y')\right)\right)\right]$$

(21)

6: **end for**
---

where $sgn$ is a sign function. Our implementation of iterative DPO is similar to REBEL where, at each iteration, we update with respect to $\pi_{\theta_t}$. The major difference is that REBEL regresses toward the differences in rewards while iterative DPO only utilizes the pairwise preference signal from the rewards.

---

[4]HuggingFace Model Card: EleutherAI/pythia-1.4b-deduped
[5]HuggingFace Model Card: vwxyzjn/EleutherAI_pythia-2.8b-deduped__sft__tldr
[6]HuggingFace Model Card: vwxyzjn/EleutherAI_pythia-6.9b-deduped__sft__tldr
[7]HuggingFace Model Card: vwxyzjn/EleutherAI_pythia-2.8b-deduped__reward__tldr
[8]HuggingFace Model Card: vwxyzjn/EleutherAI_pythia-6.9b-deduped__reward__tldr

### H.1.4 Reward Details

To ensure that $\pi_\theta$ remains close to $\pi_{\theta_0}$, we apply an additional KL penalty to the reward:

$$r(x, y) = RM(x, y) - \gamma(\ln \pi_{\theta_t}(y|x) - \ln \pi_{\theta_0}(y|x)) \tag{22}$$

where $RM(x, y)$ is score from the reward model given prompt $x$ and response $y$. Furthermore, to ensure that the generations terminate within the maximum generation length, we penalize any generation that exceeds this length by setting $r(x, y)$ to a small fixed constant, $\Gamma$.

For *TL;DR* summarization, we set $\gamma = 0.05$ and $\Gamma = -1$.

### H.1.5 Hyperparameter Details

**Parameter setting for *TL;DR* summarization**

| Setting | Parameters | |
|---|---|---|
| SFT & RM | batch size: 64
learning rate: 3e-6 | schedule: cosine decay
train epochs: 1 |
| PPO | batch size: 512
learning rate: 3e-6
schedule: linear decay
train epochs: 1
num epochs: 4 | discount factor: 1
gae $\lambda$: 0.95
clip ratio: 0.2
value function coeff: 0.1
kl coefficient: 0.05 |
| DPO | batch size: 64
learning rate: 3e-6
schedule: linear decay | train epochs: 1
$\beta$: 0.05 |
| RLOO | batch size: 512
learning rate: 3e-6
schedule: linear decay | train epochs: 1
kl coefficient: 0.05
K: 2 or 4 |
| REINFORCE | batch size: 512
learning rate: 3e-6
schedule: linear decay | train epochs: 1
kl coefficient: 0.05 |
| Iterative DPO | batch size: 512
learning rate: 3e-6
schedule: linear decay
train epochs: 1 | num epochs: 4
$\beta$: 0.05
kl coefficient: 0.05 |
| REBEL | batch size: 512
learning rate: 3e-6
schedule: linear decay
train epochs: 1 | num epochs: 4
$\eta$: 1.0
kl coefficient: 0.05 |
| LoRA Adapter Config | r: 1024
$\alpha$: 2048 | dropout: 0.0
bias: False |
| Generation | sampling: true
top k: 0.0
top p: 1.0 | min length: 53
max new tokens: 53
temperature: 0.1 |

## H.1.6 Winrate Details

We are using `gpt-4-0613` checkpoint for winrate computations. Below we show the prompt for winrate evaluation and an example evaluation from GPT4.

**Prompt for Winrate**

---

Which of the following summaries does a better job of summarizing the most important points in the given forum post, without including unimportant or irrelevant details? Judge based on accuracy, coverage, and coherence.

### Post:
{{post}}

### Summary A:
{{summarya}}

### Summary B:
{{summaryb}}

### Instructions:
FIRST provide a one-sentence comparison of the two summaries, explaining which you prefer and why. SECOND, on a new line, state only "A" or "B" to indicate your choice. Your response should use the format:
Comparison: <one-sentence comparison and explanation >
Preferred: <"A" or "B">

---

**Example Evaluation from GPT4**

| | |
|---|---|
| **Prompt** | SUBREDDIT: r/AskReddit |
| | TITLE: How do you get someone out of your head? |
| | POST: Hi,
I'm 22, and I have been with my girlfriend for 5 years now. We recently moved together. We've always loved each other intensely. |
| | Problem, I recently started to have feelings for an other person (a friend). This person has had a boyfriend for now 3 years, and has absolutely no ideas. Those feelings were so strong, it was hard to hide them. After 2 months of me being distant and really sad, my girlfriend forced me to say what was bothering me. I'm not a good liar, and now she knows. |
| | We decided to give us a week alone, I went to my parents. |
| | Now, I'm completely lost. I keep on thinking about this person, and I hate that. I would like for those feelings to go away, to leave me alone. But I can't. |
| | What do I do? It's been 3 months now, and I'm just desperate. |
| | TL;DR: |
| **Reference (Summary A)** | long relationship; fell in love with an other person; admitted it; would like it to disappear, though it doesn't. |
| REBEL **Generation (Summary B)** | I recently started to have feelings for an other person (a friend). We decided to give us a week alone, I keep on thinking about that person, and I hate it. What do I do? |
| **Evaluation from GPT4** | Comparison: Summary A is too brief and rather disjointed, while Summary B more accurately conveys the emotional conflict portrayed in the forum post in a coherent manner.
Preferred: B |

## H.2 General Chat

### H.2.1 Dataset Details

We present dataset details in Table 5.

Table 5: Dataset details for General Chat

| Dataset | Size | Prompt Length | Generation Length |
|---------|------|---------------|-------------------|
| Nectar[9] | 183k | 1024 | 1024 |
| UltraFeedback[10] | 64k | 1024 | 1024 |

### H.2.2 Model Details

For OpenChat-3.5[11], we only train the last four layers and keep other layers frozen. For Meta-Llama-3-8B-Instruct[12], we perform full-parameter training. For Starling-RM-7B-alpha[13] and ArmoRM-Llama3-8B-v0.1[14], we directly use the reward scores without any normalizations. We filter out prompts that are longer than $1,024$ tokens (2.3%) to fit the input length. OpenChat-3.5 is trained for four days, and Meta-Llama-3-8B-Instruct is train for one day on 8 H100 GPUs.

### H.2.3 Reward Details

To ensure that $\pi_\theta$ remains close to $\pi_{\theta_0}$, we apply an additional KL penalty to the reward:

$$r(x, y) = RM(x, y) - \gamma(\ln \pi_{\theta_t}(y|x) - \ln \pi_{\theta_0}(y|x)) \tag{23}$$

where $RM(x, y)$ is score from the reward model given prompt $x$ and response $y$. Furthermore, to ensure that the generations terminate within the maximum generation length, we penalize any generation that exceeds this length by setting $r(x, y)$ to a small fixed constant, $\Gamma$.

For the general chat experiments, we set $\Gamma = -4$.

### H.2.4 Hyperparameter Details

**Parameter setting for General Chat**

| Setting | Parameters |
|---------|-----------|
| Base model: OpenChat-3.5
Reward Model: Starling-RM-7B-alpha
Dataset: Nectar | batch size: 32
learning rate: 1e-7
schedule: linear decay
train epochs: 1
num epochs: 4
$\eta$: 1.0
$\gamma$: 0.05
$\Gamma$: -4 |
| Base model: Meta-Llama-3-8B-Instruct
Reward Model: ArmoRM
Dataset: UltraFeedback | mini-batch size: 128
learning rate: 3e-7
schedule: cosine decay
warm ratio: 0.1
train epochs: 1
iteration: 3
$\eta$: 1e6 (iter 1), 1e4 (iter 2), 1e2 (iter 3)
$\gamma$: 0 |

---

[9]HuggingFace Dataset Card: berkeley-nest/Nectar
[10]HuggingFace Dataset Card: openbmb/UltraFeedback
[11]HuggingFace Model Card: openchat/openchat_3.5
[12]HuggingFace Model Card: meta-llama/Meta-Llama-3-8B-Instruct
[13]HuggingFace Model Card: berkeley-nest/Starling-RM-7B-alpha
[14]HuggingFace Model Card: RLHFlow/ArmoRM-Llama3-8B-v0.1

### H.3 Image Generation

#### H.3.1 Dataset Details

Generation prompts: cat, dog, horse, monkey, rabbit, zebra, spider, bird, sheep, deer, cow, goat, lion, tiger, bear, raccoon, fox, wolf, lizard, beetle, ant, butterfly, fish, shark, whale, dolphin, squirrel, mouse, rat, snake, turtle, frog, chicken, duck, goose, bee, pig, turkey, fly, llama, camel, bat, gorilla, hedgehog, kangaroo.

#### H.3.2 Model Details

We use the latent consistency model (Luo et al., 2023) distillation of the Dreamshaper v7 model [15] for our experiments. Experiments are conducted on 4 A6000 GPUs with each run requiring 10 hours.

#### H.3.3 Hyperparameter Details

**Parameter setting for Consistency Models**

| Setting | Parameters |
|---------|-----------|
| PPO | advantage clip maximum: 10 
 batches per epoch: 10 
 clip range: 1e-4 
 learning rate: 1e-4 
 gradient accumulation steps: 8 
 max gradient norm: 5 
 number of epochs: 100 
 horizon: 8 
 number of sample inner epochs: 2 
 sample batch size (per GPU): 8 
 rolling statistics buffer size: 32 
 rolling statistics min count: 16 
 train batch size (per GPU): 2 
 LoRA rank: 8 
 Lora $\alpha$: 8 |
| REBEL | advantage clip maximum: 10 
 batches per epoch: 10 
 learning rate: 3e-4 
 $\eta$: 1 
 gradient accumulation steps: 8 
 max gradient norm: 5 
 number of epochs: 100 
 horizon: 8 
 number of sample inner epochs: 1 
 sample batch size (per GPU): 8 
 rolling statistics buffer size: 32 
 rolling statistics min count: 16 
 train batch size (per GPU): 1 
 LoRA rank: 8 
 Lora $\alpha$: 8 |

---

[15]Huggingface model card: SimianLuo/LCM_Dreamshaper_v7

# I Example Generations

## I.1 Summarization Generations from a 2.8B model

| | |
|---|---|
| **Prompt** | SUBREDDIT: r/relationships |
| | TITLE: Me [18 M] with my college friend [19 F] about 1 month, I wanna hang out with her but I don't have a car. |
| | POST: This is the link for an earlier thread I posted about her |
| | She basically doesn't have any friends and will spend all her spring break at home playing video games. I wanna hang out with her, but both of us don't have a car. I live with my grandma 10 minutes away from her house. Would it be weird to ask her to spend a day with us playing video games and eating pizza? |
| | Another thing I could do is ask her to go see a movie with me, but I would just meet her there since I'll be going with a bike. Do you guys think it's okay to do that? I also have 0 experience with girls. You can say this is the first girl in my life. |
| | TL;DR: |
| **Generation** | My college friend doesn't have any friends and spends all her spring break at home playing video games. I wanna hang out with her but neither of us have a car. Do you think it's okay to ask her to hang out? |
| **Prompt** | SUBREDDIT: r/AskReddit |
| | TITLE: Reddit, what kind of effect do you think that crime shows/movies might have in reducing the occurrence of thought-out, "intelligent" crime? |
| | POST: I had this thought a few days ago when watching an episode of NCIS and they did the usual "create a super high resolution image from a very low resolution image." I got to wondering how much of it is real and how much of it is fake (or, conversely, how much might be under-exaggerated). Based on these types of shows/movies (bank robbery, conning, etc.) where the bad guy nearly 100% of the time gets caught, only an idiot would think it would be a good idea to do any of this stuff, and from my [limited] knowledge of the general criminal, I'm under the impression that they are mainly uneducated people in dire situations (or psychopaths/mentally unstable.. but I'm thinking mostly with regards the heist type of crime – not assault or murder) that are driven to extreme lengths to make the "easy"/"quick" dollar. So I wondered, is it possible that these shows/movies that show the criminal, no matter how smart and creative, almost always getting caught could actually be (subconsciously or consciously) reducing the amount of more well thought through crime throughout the world? |
| | TL;DR: |
| **Generation** | I had this thought recently watching NCIS episode where criminals almost always get caught, is it possible that these crime shows/movies that show the criminal almost always getting caught could actually be (subconsciously or consciously) reducing the crime? |

## I.2 Chat Generations from REBEL-Llama-3

| | |
|---|---|
| **Prompt** | Hi, I'd like to play ice hockey. Can you explain how the game works? |
| **Generation** | Hockey enthusiast! I'd be happy to explain the basics of ice hockey. Here's a comprehensive overview to get you started: |

**Objective:**
The objective of ice hockey is to score more goals than the opposing team by using your stick to hit a puck (a rubber disc) into the opponent's goal.

**Gameplay:**
1. **Game Structure:** A standard ice hockey game consists of three periods, each lasting 20 minutes.
2. **Teams:** Each team consists of six players on the ice at a time:
   - **Forwards** (3): Players who score goals and create scoring opportunities.
   - **Defensemen** (2): Players who defend their goal and prevent the opposing team from scoring.
   - **Goalie** (1): The player responsible for preventing the puck from entering the goal.

3. **Gameplay Flow:** The game starts with a face-off, where two players from opposing teams try to gain control of the puck after it's dropped onto the ice by the referee. Players can move the puck using their sticks, skates, or by carrying it.
4. **Scoring:** A goal is scored when a player hits the puck into the opponent's goal using their stick, skate, or body.
5. **Penalties:** If a player breaks the rules, they can be sent to the penalty box for a set amount of time (2-5 minutes). During this time, their team must play shorthanded (with one fewer player on the ice).
6. **Power Play:** If the opposing team scores during a penalty, the penalized team gets a power play, where they have a man advantage on the ice.
7. **Icing:** If a player shoots the puck from behind their own blue line and it crosses the opponent's goal line without being touched, it's considered icing, and the opposing team is awarded a face-off in their own end zone.
8. **Offside:** A player cannot enter the opponent's zone before the puck, or they'll be called offside.

**Common Penalties:**
- **Hooking:** Using the stick to hook an opponent's leg or body.
- **Holding:** Grabbing onto another player's jersey or equipment.
- **Tripping:** Using a stick, leg, or body to trip an opponent.
- **Interference:** Interfering with an opponent who doesn't have possession of the puck.

**Equipment:**
- **Stick:** A long, flat stick used to pass, shoot, and control the puck.
- **Skates:** Specialized ice skates designed for speed, agility, and stopping.
- **Pads:** Protective gear worn by players, including shoulder pads, elbow pads, gloves, shin guards, and hockey pants.
- **Helmet:** A protective helmet worn by all players to prevent head injuries.
- **Goalie Gear:** Specialized equipment worn by goalies, including a large pad covering their body, gloves, and a mask.

**Basic Strategies:**
- **Forechecking:** Applying pressure to the opponent's defensemen to gain possession of the puck.
- **Backchecking:** Pressuring the opponent's forwards to prevent them from scoring.
- **Breakouts:** Quickly moving the puck from your own end zone to the neutral zone to create scoring opportunities.

This is a solid foundation to get you started. As you play more, you'll learn more about the game's nuances and strategies. Have fun playing ice hockey!

| Prompt | Why do people give Reddit Gold to admins? |
|---|---|
| Generation | On Reddit, Reddit Gold (also known as Ğold) is a virtual currency that allows users to show appreciation for content creators, moderators, and other users who contribute positively to the community. When someone gives Reddit Gold to an admin (or moderator), it's often a way to express gratitude for their efforts in maintaining the community's quality, resolving issues, and keeping the discussion civil. |

Here are some reasons why people might give Reddit Gold to admins:
1. **Appreciation for moderation**: Admins work tirelessly to ensure that discussions remain on-topic, respectful, and follow community guidelines. Giving Gold acknowledges their dedication and hard work in maintaining a positive environment.
2. **Recognition of expertise**: Admins often possess in-depth knowledge about specific topics or communities. Giving Gold shows appreciation for their expertise and willingness to share it with others.
3. **Support for community building**: Admins play a crucial role in fostering engagement, encouraging participation, and creating opportunities for users to connect with each other. Gold gifts demonstrate support for their efforts in building a strong community.
4. **Rewarding good behavior**: When admins demonstrate excellent behavior, such as resolving conflicts fairly or providing helpful guidance, users might give Gold as a way to recognize and reward their good behavior.
5. **Boosting morale**: Giving Gold can be a morale booster for admins, acknowledging their value to the community and motivating them to continue their excellent work.

It's worth noting that Reddit Gold is not required or expected, and admins don't receive any direct benefits from receiving Gold. However, the gesture of appreciation can go a long way in fostering a positive and supportive community.

## I.3   Image Generations

Example Generations of REBEL

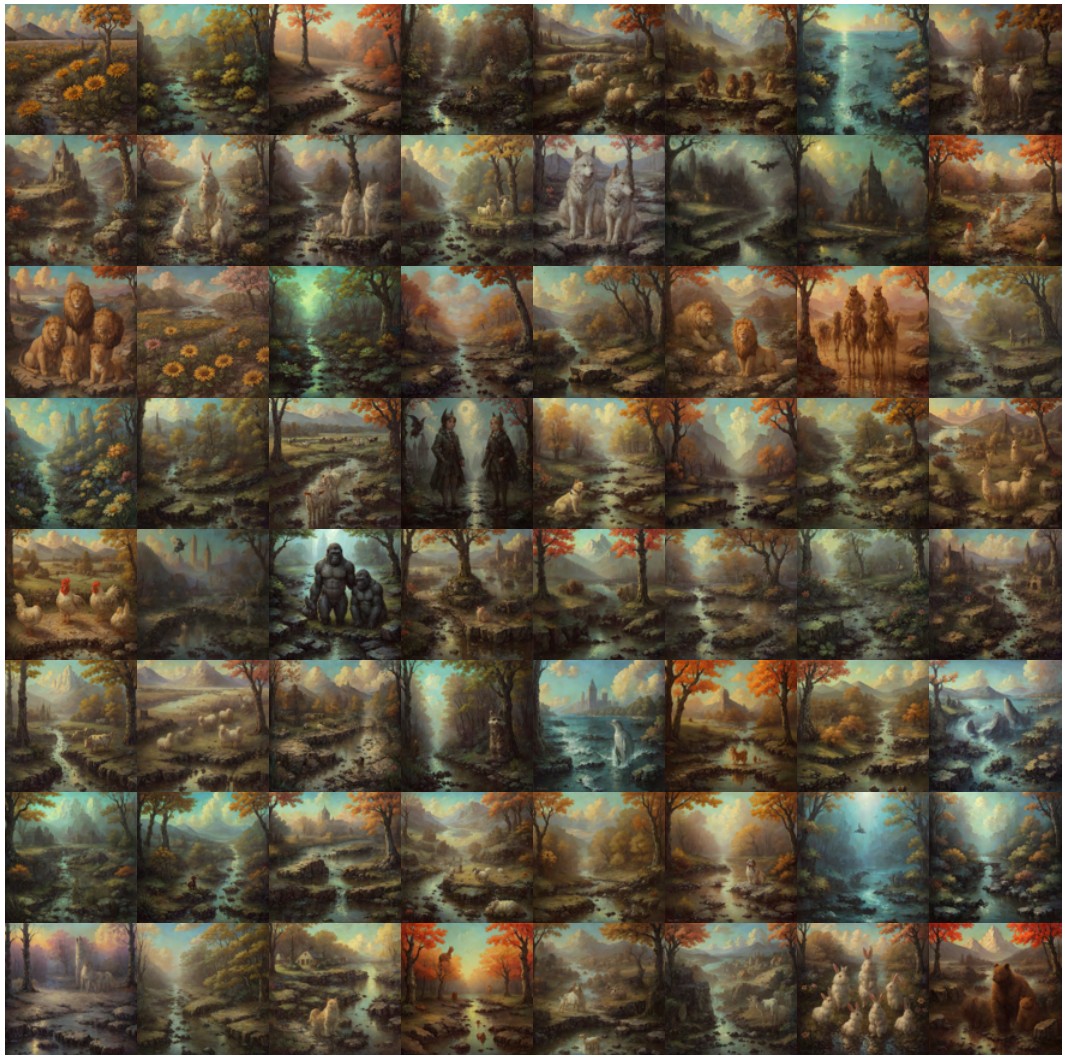

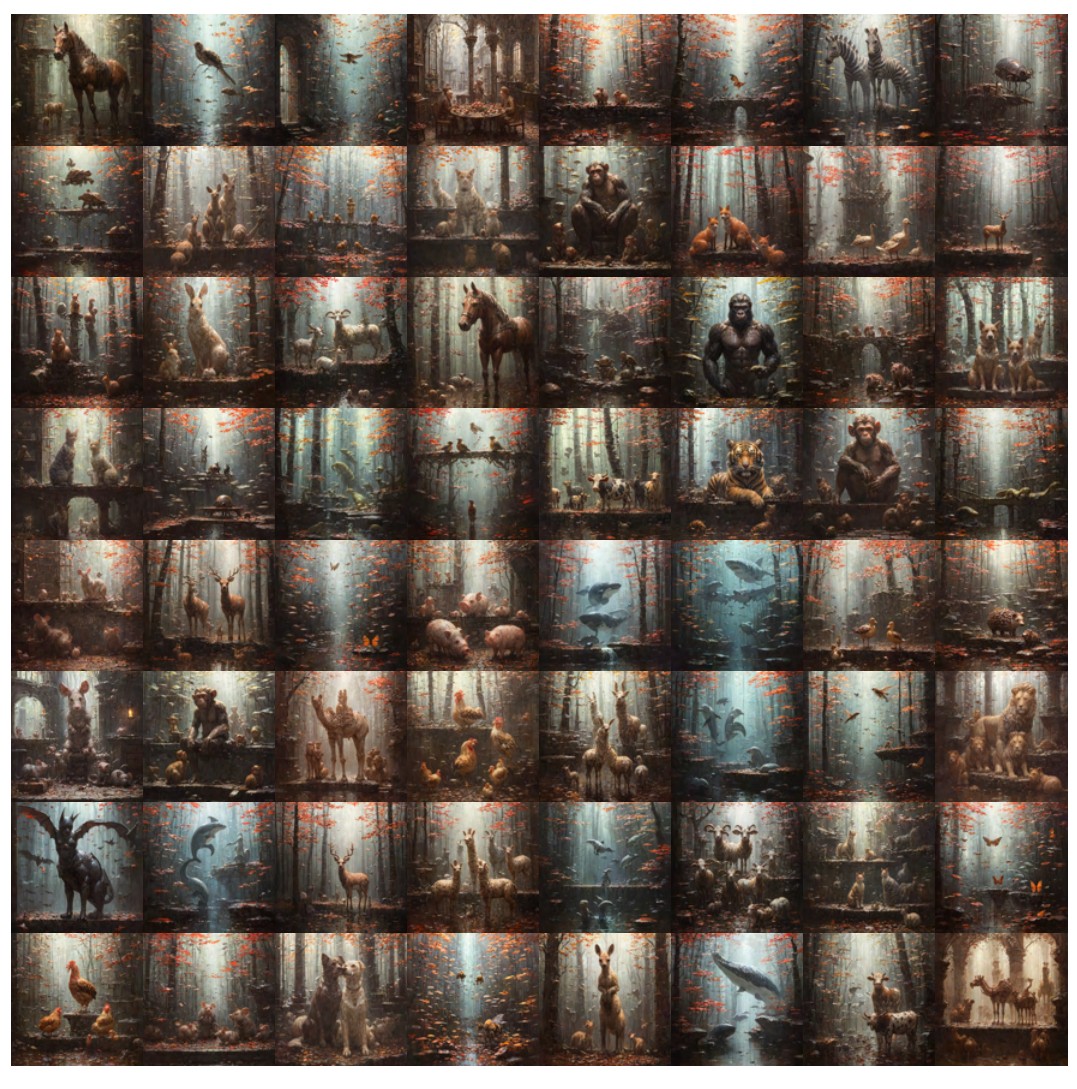

## J Ablation Analysis

| $\eta$ | Winrate ($\uparrow$) | RM Score ($\uparrow$) | KL$(\pi\|\pi_{ref})$ ($\downarrow$) |
|---|---|---|---|
| 0.3 | 55.5% | 1.37 | 10.4 |
| 0.7 | 59.9% | 1.60 | 14.2 |
| 1.0 | 70.2% | 2.44 | 29.0 |
| 2.0 | 62.5% | 1.76 | 16.9 |

Table 6: REBEL ablation of the key hyperparameter $\eta$ on summarization task and 2.8B model. The best-performing $\eta$ for each metric is highlighted in bold.

Just like DPO, tuning REBEL is much more straightforward than PPO since the only hyperparameter REBEL introduced is $\eta$. We investigate how sensitive REBEL is to learning rate $\eta$ in the loss. The results of ablation on summarization task and 2.8B model is shown in Table 6 with the same setting detailed in Appendix H.1.5 except for $\eta$. REBEL achieves the best performance when $\eta = 1$, while increasing or decreasing $\eta$ leads to decreased performance. Our result here indicates that $\eta$ is an important hyperparameter that requires tuning for achieving a good performance. Setting $\eta$ to 1.0 is a good starting point since, for all of our experiments from language modeling to image generation, $\eta = 1$ achieves the best results.

## K Trade-off between Reward Model Score and KL-divergence

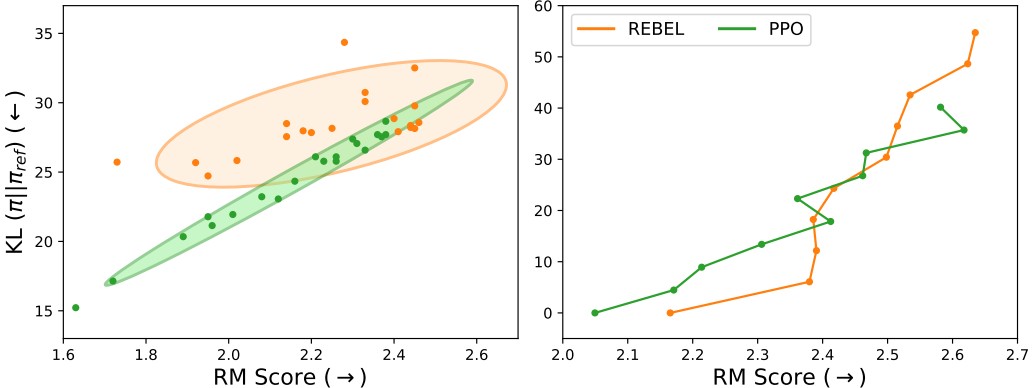

Figure 5: Plot of Reward vs KL-Divergence for 2.8B REBEL and PPO for summarization. We evaluate the models across the entire test set every 100 steps for 2,000 steps. Left: each point represents the average reward score and KL-divergence for a specific time step; the eclipse represents the confidence interval with 2 standard deviations. Right: we divide the KL distribution at the 2,000-step into 10 bins with equal size and average the corresponding RM scores in each bin.

The trade-off between the reward model score and KL-divergence is shown in Figure 5. We evaluate the 2.8B REBEL and PPO every 400 gradient updates during training for 8,000 updates on summarization. The sample complexity of each update is held constant across both algorithms for fair comparison. For the left plot, each point represents the average divergence and score over the entire test set, and the eclipse represents the confidence interval with 2 standard deviations. As observed previously, PPO exhibits lower divergence, whereas REBEL shows higher divergence but is capable of achieving larger RM scores. Notably, towards the end of the training (going to the right part of the left plot), REBEL and PPO have similar KL and RM scores. For the right plot in Figure 5, we analyze a single checkpoint for each algorithm at the end of training. For each algorithm, we group every generation from the test set by its KL distribution into 10 equally sized bins and calculate the average of the corresponding RM score for each bin. We can see that REBEL achieves higher RM scores for generations with small divergence while requiring larger divergence for generations with the highest scores.

## L Regression Loss During Training

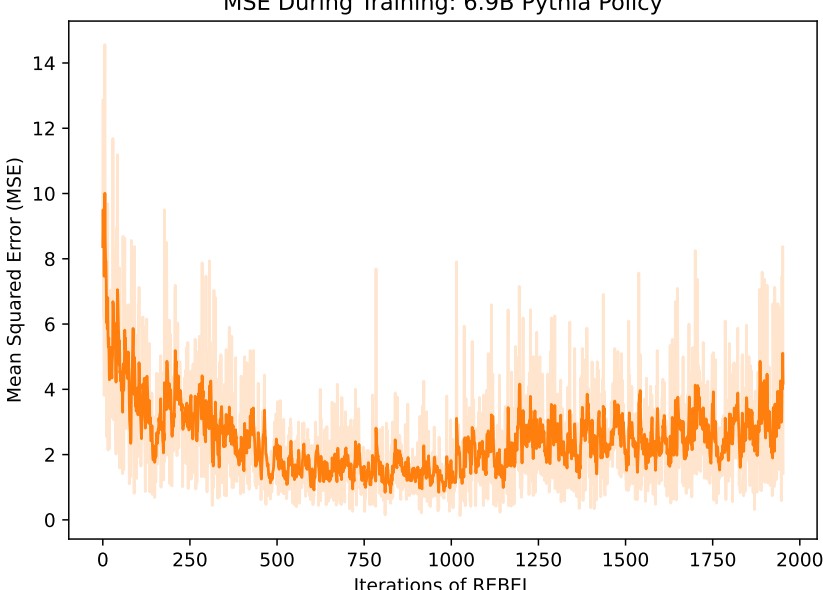

Figure 6: REBEL's reward difference prediction error throughout training of our 6.9B parameter policy on the summarization task. The reward used for this task is unbounded with the range of values of the human labels in the validation set being $[-6.81, 7.31]$. We plot both the smoothed values with a moving average and the loss vales at each iteration.

Figure 6 shows the observed loss of Eq. 1 that we observed when finetuning the 6.9B Pythia model on summarization. We see that REBEL minimizes the loss throughout training maintaining a relatively low mean squared error given that our observed rewards were mostly between $[-10, 10]$. Note that our learned reward model, however, is unbounded.

## M Breakdown of MT-Bench

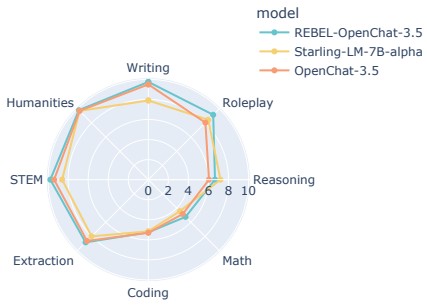

Figure 7: Breakdown of MT-Bench results over eight dimensions.

Figure 7 shows the breakdown of MT-Bench results. REBEL (REBEL-OpenChat-3.5) outperforms both APA (Starling-LM-7B-alpha) and base (OpenChat-3.5) models on six out of eight dimensions including writing, roleplay, math, extraction, STEM, and humanities.

