# OpenReview forum: "REBEL: Reinforcement Learning via Regressing Relative Rewards"
_NeurIPS.cc/2024/Conference — NeurIPS 2024 poster_

### Official Review · Reviewer_zZ2j · 2024-07-09

**Soundness:** 3
**Presentation:** 3
**Contribution:** 3
**Rating:** 7
**Confidence:** 3

**Summary:**

This paper reduces the complex policy optimization procedure of alignment to a simple regression objective, using the relation between optimal policy and reward. The paper conduct detailed theoretical analysis in revealing the relation between the proposed algorithm *REBEL* and *NPG/MD*. Comprehensive experiments in both text and image generation exhibit the effectiveness of *REBEL*.

**Strengths:**

1. This paper studies simplified version of policy optimization in RLHF (compared to PPO), which is a research topic of interest.
2. The theoretical analysis of *REBEL* is detailed and insightful.
3. The presentation of this paper is logically clear and has good readability.
4. The experiments in this paper are comprehensive, and the experimental results are well presented.

**Weaknesses:**

1. The statement "REBEL ... be extended to handle intransistive preferences ...." in the abstract is not adequately presented in the main content of the paper. As the major influence brought by intransistive preferences is the degradation of reward score accuracy, which is not addressed by this paper.
2. I would suggest the authors to summarize the limitations of the proposed method in a separate "Limitations" section.

**Questions:**

none

---

> ### Author Rebuttal · Authors · 2024-08-06
>
> Thank you for your valuable review of our paper.
>
> > The statement "REBEL ... be extended to handle intransitive preferences ...." in the abstract is not adequately presented in the main content of the paper. As the major influence brought by intransistive preferences is the degradation of reward score accuracy, which is not addressed by this paper.
>
> We appreciate your observation and agree with your assessment. Since our focus is not on the preference model itself, we did not conduct experiments specifically targeting preference models. However, we provided a method to extend REBEL to preference models. The extension of REBEL to address intransitive preferences is discussed in detail in Appendix D, with further analysis provided in Appendix G. How to pre-train a good preference model without any degradation of reward score accuracy is beyond the scope of our paper.
>
> > I would suggest the authors to summarize the limitations of the proposed method in a separate "Limitations" section.
>
> Thank you for the suggestion. We will include a dedicated "Limitations" section in the next version of the paper.

---

> > ### Comment · Reviewer_zZ2j · 2024-08-13
> >
> > Thank you for your response, my concerns have been addressed during the rebuttal and I decide to keep my score.

---

### Official Review · Reviewer_ihUH · 2024-07-16

**Soundness:** 3
**Presentation:** 3
**Contribution:** 2
**Rating:** 5
**Confidence:** 4

**Summary:**

This paper proposes the REBEL algorithm that reduces policy optimization to iteratively solving squared loss regression problems on the difference in rewards between trajectories, based on DPO's analysis. The paper transforms the resulting equation for r(x, y) presented in DPO to a regression loss function, and avoids the intractable calculation of Z(x) by calculating the loss based on a pair of samples from the same input prompt x, i.e., (x, y) and (x, y'). One of the goals for REBEL is to serve as a simple and lightweight RL algorithm that eliminates the need for complex components like value functions and clipping heuristics used in PPO. The authors provide a theoretical analysis showing that Natural Policy Gradient can be seen as a special case of REBEL under some assumptions. The authors conduct two kinds of empirical analysis including language modeling and image generation tasks to demonstrate the performance of REBEL.

**Strengths:**

- Originality:
    - This paper presents a new angle by transforming the analysis of the reward function presented in the DPO paper into a reward regression loss, leading to the proposed REBEL algorithm.
    - The authors make connections between REBEL and existing RL methods like NPG considering some assumptions, showing that these algorithms can be seen as special cases or approximations of REBEL under certain conditions.

- Quality:
    - The paper provides a thorough theoretical analysis comparing REBEL with existing RL approaches.

- Clarity:
    - The paper is well-written and easy to understand, with a clear logical flow from motivation to theoretical analysis to empirical validation. The authors do an good job of explaining the intuition behind REBEL and highlighting its connections to prior work.

- Significance:
    - The paper tackles the important problem of developing simpler and more efficient RL algorithms that can scale to large-scale generative model fine-tuning.

**Weaknesses:**

1. Insufficient experimental validation and limited baseline comparisons:
- While the paper presents empirical results on language modeling and image generation tasks, the experimental validation of REBEL could be more comprehensive. The authors should consider including a wider range of benchmarks and datasets to demonstrate the generality and robustness of their approach.
- The comparison with baseline algorithms like PPO and DPO is somewhat limited. The authors should provide more details on the hyperparameter settings and training procedures for the baselines to ensure a fair comparison. Moreover, the poor performance of DPO compared to PPO in the experiments raises questions about the implementation or hyperparameter choices.
- The authors claim that REBEL matches the strongest known theoretical guarantees in terms of convergence and sample complexity. However, the experiments only compare performance at a specific epoch without demonstrating improved sample efficiency. Convergence plots showing the performance of REBEL and baselines over the course of training would provide a clearer picture of the sample efficiency and convergence properties.

2. Lack of support for certain claims and limited exploration of key aspects:
- The paper makes several claims regarding the advantages of REBEL, such as its ability to handle intransitive preferences, incorporate offline datasets, and apply to deterministic MDPs. However, there is a lack of corresponding experimental evidence or theoretical analysis to substantiate these claims.
- The relationship between the regressor's performance and the quality of the dataset used for training is not explored in depth. Insights or experiments that investigate how dataset quality and diversity affect the regressor's ability to capture an improved policy would strengthen the paper.
- The choice of base distribution \mu is mentioned as a determining factor for whether REBEL is hybrid or fully online. However, the paper does not provide experimental results comparing different forms of \mu across various tasks or practical guidelines for choosing \mu in real-world applications.

3. Inconsistencies and potential conflicts with previous statements:
- The authors mention that critic-based variance reduction might be necessary for high-variance trajectory-level rewards in stochastic MDPs, which seems to contradict the criticism of PPO's complexity in the introductory section. The lack of experimental support for REBEL's performance in stochastic MDPs is a significant limitation, and the authors should provide preliminary results or theoretical insights to support their claims.

**Questions:**

1. Sample efficiency and convergence guarantees:
- The authors claim that REBEL matches the strongest known theoretical guarantees in terms of convergence and sample complexity. However, the experiments only compare performance at a specific epoch without demonstrating improved sample efficiency. Can the authors provide experimental results that support their claim of improved sample efficiency compared to other algorithms?
- It would be helpful to see convergence plots that show the performance of REBEL and baseline algorithms over the course of training, rather than just at a selected epoch. This would provide a clearer picture of the sample efficiency and convergence properties of REBEL.
2. Relationship between regressor performance and dataset quality:
- The authors state that a regressor that can predict the difference in rewards between trajectories implicitly captures an improved policy. Is the performance of this regressor dependent on the quality of the dataset used for training? How does the quality of the dataset affect the regressor's ability to capture an improved policy?
- Can the authors provide insights or experiments that explore the relationship between dataset quality and the effectiveness of REBEL?
3. Applicability to deterministic MDPs:
- The authors mention that REBEL can be applied to any deterministic MDP where the initial state is x and the trajectory y consists of a sequence of actions. Is there any experimental or theoretical support for this claim?
- It would strengthen the paper if the authors could provide empirical results or theoretical analysis that demonstrates the effectiveness of REBEL in deterministic MDPs beyond the bandit formulation.
4. Choice of base distribution \mu:
- The authors state that the choice of base distribution \mu determines whether REBEL is hybrid or fully online. Can they provide experimental results that compare different forms of \mu across various types of tasks? What are the practical guidelines for choosing \mu in real-world applications?
- Insights into the impact of different choices of \mu on the performance and behavior of REBEL would be valuable for practitioners looking to apply this algorithm.
5. Stochastic MDPs and the need for critic-based variance reduction:
- The authors leave the experimental validation of REBEL in stochastic MDPs for future work but mention that trajectory-level rewards can be high-variance, potentially requiring critic-based variance reduction. In what practical situations would the transition dynamics be stochastic? If critic-based variance reduction is needed, how does this align with the introductory section's criticism of PPO's complexity?
- The lack of experimental support for REBEL's performance in stochastic MDPs is a significant limitation. Can the authors provide any preliminary results or theoretical insights that support their claims about REBEL's applicability to stochastic environments?
6. Performance comparison with baselines:
- In the experiments conducted by the authors, DPO performs significantly worse than PPO, especially in Table 1, where DPO is inferior in every case. Can the authors provide an explanation for this discrepancy? Is it due to differences in implementation or hyperparameter settings?
- In Figure 3, the comparison between PPO and REBEL is made at an intermediate checkpoint where REBEL observes a higher reward under the reward model. Is it possible that PPO has already overfit at this selected epoch? How was this specific epoch number chosen for REBEL? What would the comparison look like if the best-performing epoch for each algorithm were considered? Additionally, why is the comparison limited to only PPO? It would be informative to include other state-of-the-art RL algorithms in the comparison to better understand the relative performance of REBEL.

**Limitations:**

Yes.

---

> ### Author Rebuttal · Authors · 2024-08-06
>
> Thank you for your constructive feedback. We address each of your points below.
>
> > Insufficient experimental validation and limited baseline comparisons
>
> > Performance comparison with baselines
>
> Our experimental section is comprehensive compared to previous works on RLHF [1, 2], incorporating a general chat dataset and an image generation task to evaluate the robustness and generality of REBEL. We compared REBEL with several algorithms, including DPO, PPO, iterative DPO, REINFORCE, RLOO, and APA. Detailed hyperparameter settings are provided in Appendix H.1.4, H.2.3, H.3.3.
>
> The lower performance of DPO compared to PPO is not due to implementation or hyperparameter choices. On TL;DR, our DPO results are better than the ones reported in [3]. Results for PPO, REINFORCE, and RLOO are directly obtained from prior papers [2, 4] which are exclusively focusing on these algorithms on TL;DR. For general chat, we directly report winrates from the released starling-alpha model by APA's authors. Thus, we believe that our comparison to baselines is fair.
>
> In Figure 3, the comparison at an intermediate checkpoint is intended to highlight the sample efficiency of REBEL. There is no indication that PPO has overfitted at this epoch, as the image quality of PPO continues to improve afterward.
>
> > Sample efficiency and convergence guarantees
>
> For the image generation task, as illustrated in Figure 4, REBEL converges faster during the initial training phase and eventually achieves performance comparable to that of PPO. In addition, we plot the reward vs. step for TL;DR dataset in the pdf of the global rebuttal. The plot demonstrates REBEL's faster convergence compared to iterative DPO and PPO.
>
> > Lack of support for certain claims and limited exploration of key aspects
>
> > Applicability to deterministic MDPs
>
> > Choice of base distribution \mu
>
> Since our focus is not on the preference model itself, we did not conduct experiments specifically targeting preference models. However, we provided a method to extend REBEL to preference models in Appendix D, with further analysis provided in Appendix G.
>
> In our experiments, we found that setting $\mu=\pi_t$ yielded better results. We attribute this to the lower quality of the offline dataset. In TL;DR summarization and Ultrafeedback, our trained policies can generate better responses than the ones in the datasets. This is shown in Table 1, where the 2.8B and 6.9B models can easily reach high winrates compared to offline human demonstrations. In this case, setting $\mu$ to $\pi_{ref}$ or the offline data does not help significantly. Therefore, we use $\pi_t$ as $\mu$ in all experiments.
>
> Since transitions are deterministic, we could formulate this as a bandit problem where $x$ is the initial state and $y$ is the trajectory consisting of a sequence of actions [1, 2, 10]. Applying token generations in LLMs and image generation to deterministic MDPs has been proposed in prior RLHF works [5, 6, 7, 8]. We adapt this bandit setup to make a fair comparison to the previous works.
>
> > Inconsistencies and potential conflicts with previous statements
>
> > Stochastic MDPs and the need for critic-based variance reduction
>
> Lack of experimental support in stochastic MDPs is not a significant limitation as deterministic MDPs have wide applications in LLMs and image generations, which are high-impact areas. We emphasize that all prior RLHF works [1, 2, 4, 6] all focus on deterministic MDP.
>
> There is also no conflict in our criticism of PPO's complexity. Our paper focuses on deterministic transitions; in this context, many prior work have argued that PPO and critic-based variance reduction methods are not necessary [2, 8, 9]. We agree that in highly stochastic settings, critics might still be necessary, which is indeed stated in the paper.
>
> > Relationship between regressor performance and dataset quality:
>
> A high-quality dataset, i.e. a dataset that has diverse coverage, would increase the generalization ability of the learned regressor. E.g., if the training distribution is diverse, then the learned regressor can achieve better performance under the comparator policy $\pi^*$ distributions, which, as our theory indicates, will ensure convergence to $\pi^*$. Our work focuses on online RL and studying the dataset quality is not the main focus of the paper.
>
> [1] Rafailov R, Sharma A, Mitchell E, Manning CD, Ermon S, Finn C. Direct preference optimization: Your language model is secretly a reward model.
>
> [2] Ahmadian A, Cremer C, Gallé M, Fadaee M, Kreutzer J, Üstün A, Hooker S. Back to basics: Revisiting reinforce style optimization for learning from human feedback in llms.
>
> [3] Rafailov R, Chittepu Y, Park R, Sikchi H, Hejna J, Knox B, Finn C, Niekum S. Scaling laws for reward model overoptimization in direct alignment algorithms.
>
> [4] Huang S, Noukhovitch M, Hosseini A, Rasul K, Wang W, Tunstall L. The N+ Implementation Details of RLHF with PPO: A Case Study on TL; DR Summarization.
>
> [5] Ramamurthy R, Ammanabrolu P, Brantley K, Hessel J, Sifa R, Bauckhage C, Hajishirzi H, Choi Y. Is Reinforcement Learning (Not) for Natural Language Processing: Benchmarks, Baselines, and Building Blocks for Natural Language Policy Optimization.
>
> [6] Stiennon N, Ouyang L, Wu J, Ziegler D, Lowe R, Voss C, Radford A, Amodei D, Christiano PF. Learning to summarize with human feedback.
>
> [7] Chang JD, Shan W, Oertell O, Brantley K, Misra D, Lee JD, Sun W. Dataset reset policy optimization for rlhf.
>
> [8] Black K, Janner M, Du Y, Kostrikov I, Levine S. Training diffusion models with reinforcement learning.
>
> [9] Oertell O, Chang JD, Zhang Y, Brantley K, Sun W. Rl for consistency models: Faster reward guided text-to-image generation.
>
> [10] Wu T, Zhu B, Zhang R, Wen Z, Ramchandran K, Jiao J. Pairwise proximal policy optimization: Harnessing relative feedback for llm alignment.

---

> > ### Comment · Reviewer_ihUH · 2024-08-12
> >
> > Thank the authors for the rebuttal. However, I still find some of the responses to my questions to be somewhat evasive, and I would appreciate more detailed explanations from the authors. I would also be happy to increase the score if all my concerns are adequately addressed.
> >
> > 1. Regarding DPO's performance:
> > The authors mention that DPO performs better than reported in [3] for the TL;DR task. However, why is this improvement limited to the TL;DR task and not observed in other tasks? Could you provide a more comprehensive explanation for DPO's underwhelming performance in other areas?
> >
> > 2. On REBEL's convergence guarantee:
> > While the plots empirically demonstrates REBEL's faster convergence, the paper's main argument centers on theoretical explanations. Could you provide a corresponding theoretical justification for this faster convergence to complement the empirical evidence?
> >
> > 3. Concerning stochastic MDPs:
> > Given the limited analysis provided for stochastic MDPs, I'm curious about the rationale for including this section in the main text, since there are not sufficient empirical support to this added part.

---

> > > ### Author Response · Authors · 2024-08-13
> > >
> > > We thank the reviewer for the response and address each of the points below.
> > >
> > > > Regarding DPO's performance: The authors mention that DPO performs better than reported in [3] for the TL;DR task. However, why is this improvement limited to the TL;DR task and not observed in other tasks? Could you provide a more comprehensive explanation for DPO's underwhelming performance in other areas?
> > >
> > > In our experiments, we only performed experiments with DPO on the TL;DR summarization task. The improvement in performance regarding summarization could be due to differences in prompt, test time model temperature, or other factors. In particular, [3] used a temperature of 1.0 during the evaluation, while we used a temperature of 0.9. We referenced [3] to demonstrate that our DPO results are consistent with those reported by other independent researchers, indicating that our findings of DPO are reasonable within the existing literature.
> > >
> > > > On REBEL's convergence guarantee: While the plots empirically demonstrates REBEL's faster convergence, the paper's main argument centers on theoretical explanations. Could you provide a corresponding theoretical justification for this faster convergence to complement the empirical evidence?
> > >
> > > We provide a detailed analysis of REBEL's convergence rate in lines 140-147 and 200-217 of our paper. Specifically, we show that REBEL achieves a fast $1/T$ convergence under the assumption that the least square regression optimization returns the exact Bayes optimal solution. We then relax this assumption using a regression generalization bound, resulting in an agnostic regret bound with a convergence rate of $1/\sqrt{T}$. This agnostic regret bound represents the strongest type of agnostic learning results known in the RL literature.
> > >
> > > On the other hand, to the best of our knowledge, we do not know if PPO — our baseline in the image generative model experiment, has provable convergence guarantees. One reason that PPO convergence can be slower than REBEL is that PPO uses clipping to approximately maintain conservative policy update and the clipping operator throws away a non-trivial amount of training data. REBEL on the other hand does not use clipping and thus does not waste any training data.
> > >
> > > > Concerning stochastic MDPs: Given the limited analysis provided for stochastic MDPs, I'm curious about the rationale for including this section in the main text, since there are not sufficient empirical support to this added part.
> > >
> > > This point was also raised by Reviewer HFsB. We fully agree with this feedback and will move this section to the appendix.

---

### Official Review · Reviewer_HFsB · 2024-07-16

**Soundness:** 3
**Presentation:** 3
**Contribution:** 3
**Rating:** 7
**Confidence:** 3

**Summary:**

This work presents REBEL, a minimalist reinforcement learning algorithm that does policy optimization by solving a sequence of regression problems using relative rewards as targets. Theoretical analysis shows that Natural Policy Gradient (NPG) is a variant of REBEL, and thus theoretical guarantees for NPG can be applied to REBEL.  Experimental results  show that REBEL matches or outperforms existing baselines, most notably PPO and RLOO, on multiple tasks.

**Strengths:**

- The paper is well-organized and technically sound. The general flow of the paper is smooth and proposed methods are explained adequately. The paper has an appropriate number of citations and properly details existing work in the related work section.
- The method is simple to implement and has little engineering overhead. Given the minimalist implementation, the results are impressive, surpassing even PPO, which typically requires significant engineering.

**Weaknesses:**

- There are no significant weaknesses in this work, barring some clarifying details.
- I believe that at least a brief section on related work should be included in the main paper, the in-depth one can be deferred to the appendix. In terms of space, I personally do not think Section 2.2 adds much value to the main paper.

**Questions:**

- The reward model becomes increasingly off-distribution as the policy is updated. Although it is standard practice to keep reward models fixed even with iterative methods, prior works generally use it to generate preference labels between pairs of outputs. Since this work uses the difference of scores as the regression target, the off-distribution reward scores might have a greater impact here. Concisely, how significant a problem is reward model over-optimization [1] for REBEL?
-  It would be interesting to see and understand the differences between reward-weighted regression baseline (RWR) and REBEL as they have some close connections.
-  Is there an optimal choice of $\mu$ ? What are the intuitive differences between using the  $\mu= \pi_{ref}$  and $\mu= \pi_{t}$ ? As the policy improves, samples $y,y’ \sim \pi_{t}$ are in the high reward region, and it can be difficult to separate them since these might be off-distribution for the reward model. Given these constraints of the reward model, there might be better choices of $\mu$ that allow for better prediction of score differences. It would be interesting to see an ablation study on this, or a well-reasoned answer that explains the tradeoffs between different choices of $\mu$.
- Why are datasets not aggregated? Instead, only the most recently collected dataset is used for training.

[1] : Gao, L., Schulman, J., & Hilton, J. (2023, July). Scaling laws for reward model overoptimization. In International Conference on Machine Learning (pp. 10835-10866). PMLR.

---

> ### Author Rebuttal · Authors · 2024-08-06
>
> Thank you for your valuable review of our paper. We respond to your individual questions below.
>
> > I believe that at least a brief section on related work should be included in the main paper, the in-depth one can be deferred to the appendix. In terms of space, I personally do not think Section 2.2 adds much value to the main paper.
>
> Thank you for your suggestion. We will include a brief section on related work in the main paper and have a detailed discussion in the appendix.
>
> > How significant a problem is reward model over-optimization [1] for REBEL?
>
> In the context of RLHF for LLMs, we choose reward models that are highly ranked on the Reward Bench [2]. These reward models are trained using extensive datasets that include responses generated from a variety of different policies [3]. This diversity makes it challenging to over-optimize a reward model, even for methods such as REBEL. In addition, following previous work [4], we apply an addition KL penalty to the reward, $r(x, y) = RM(x, y) - \gamma (\ln \pi_{\theta_{t}}(y|x) - \ln \pi_{\theta_{0}} (y|x))$, to prevent over-optimization for both REBEL and the baseline methods.
>
> In the image generation setting, following previous work [5, 6], no KL regularization is used. We observe that both PPO and REBEL over-optimize the reward model towards the end of the training. They tend to ignore the input prompt and generate the same image that maximizes the reward model score. For fair comparison to [6], we also did not include KL.
>
> > It would be interesting to see and understand the differences between reward-weighted regression baseline (RWR) and REBEL as they have some close connections.
>
> RWR can be understood as reward-weighted imitation learning. Prior work on RL for diffusion models shows that RWR is not as effective as RL (PPO in that case) [5].  REBEL, on the other hand, is a full RL algorithm from its connection to NPG where it generalizes NPG.
>
> > Is there an optimal choice of $\mu$? What are the intuitive differences between using the $\mu=\pi_{ref}$ and $\mu=\pi_{t}$?.
>
> Setting $\mu$ to $\pi_{ref}$ or another distribution can encourage exploration, especially when the policy is still learning and the quality of the generations is not yet optimal. This approach can help in discovering diverse and potentially higher-reward samples that the current policy might not generate.
>
> In our experiments, we found that setting $\mu=\pi_t$ yielded better results. We attribute this to the lower quality of the offline dataset. In TL;DR summarization and Ultrafeedback, our trained policies can actually generate better responses than the ones in the datasets. This is shown in Table 1, where the 2.8B and 6.9B models can easily reach high winrates compared to offline human demonstrations. In this case, setting $\mu$ to $\pi_{ref}$ or the offline data does not provide significant benefits. Therefore, we use $\pi_t$ as $\mu$ in all our experiments.
>
> > Why are datasets not aggregated? Instead, only the most recently collected dataset is used for training.
>
> Our approach can be understood as online RL, e.g. PPO, where only on-policy data is used for update. We could aggregate the datasets which makes the approach more off-policy. While we have not tested this in our experiments, it could be an interesting direction to explore if being off-policy would lead to better sample efficiency.
>
> In addition, previous RLHF works [4, 8, 9] also use batches collected by the current policy (i.e. online batch), with each batch containing a new set of prompts. To ensure a fair comparison with previous methods, we primarily focus on the on-policy approach.
>
> [1] Gao, L., Schulman, J., & Hilton, J. (2023, July). Scaling laws for reward model overoptimization. In International Conference on Machine Learning (pp. 10835-10866). PMLR.
>
> [2] Lambert N, Pyatkin V, Morrison J, Miranda LJ, Lin BY, Chandu K, Dziri N, Kumar S, Zick T, Choi Y, Smith NA. Rewardbench: Evaluating reward models for language modeling. arXiv
>
> [3] Xiong, W., Dong, H., Ye, C., Wang, Z., Zhong, H., Ji, H., Jiang, N., Zhang, T. Iterative Preference Learning from Human Feedback: Bridging Theory and Practice for RLHF under KL-Constraint. 2024. arXiv
>
> [4] Huang S, Noukhovitch M, Hosseini A, Rasul K, Wang W, Tunstall L. The N+ Implementation Details of RLHF with PPO: A Case Study on TL; DR Summarization.
>
> [5] Black K, Janner M, Du Y, Kostrikov I, Levine S. Training diffusion models with reinforcement learning. arXiv
>
> [6] Oertell O, Chang JD, Zhang Y, Brantley K, Sun W. Rl for consistency models: Faster reward guided text-to-image generation. arXiv
>
> [7[ Korbak T, Shi K, Chen A, Bhalerao RV, Buckley C, Phang J, Bowman SR, Perez E. Pretraining language models with human preferences. In International Conference on Machine Learning 2023 Jul 3 (pp. 17506-17533). PMLR
>
> [8] Stiennon N, Ouyang L, Wu J, Ziegler D, Lowe R, Voss C, Radford A, Amodei D, Christiano PF. Learning to summarize with human feedback. Advances in Neural Information Processing Systems. 2020;33:3008-21.
>
> [9] Ahmadian A, Cremer C, Gallé M, Fadaee M, Kreutzer J, Üstün A, Hooker S. Back to basics: Revisiting reinforce style optimization for learning from human feedback in llms.

---

> > ### Comment · Reviewer_HFsB · 2024-08-10
> > **Reply to rebuttal**
> >
> > I thank the authors for the rebuttal. My doubts have been cleared and I have raised my score to reflect the same.

---

### Official Review · Reviewer_dEG5 · 2024-07-20

**Soundness:** 3
**Presentation:** 3
**Contribution:** 3
**Rating:** 8
**Confidence:** 3

**Summary:**

The authors present REBEL, a method for solving contextual bandit problems (such as the alignment of language models) via regressing relative rewards. They first derive their objective by demonstrating that the use of paired responses means that you can get rid of the partition function, which is impossible to estimate.

They then connect their method to previous methods in RL including detailing, but not . They demonstrate that under strong assumptions REBEL is equivalent to mirror descent, and that under assumptions of coverage by the reference policy, that REBEL produces returns close to an optimal policy.

Finally the authors run experiments on summarisation, general chat and image alignment, demonstrating their method compares favourably to other methods.

**Strengths:**

* The idea of using relative rewards to remove the partition function is a nice and simple idea
* The theoretical connections of their method to prior methods grounds their work nicely in existing RL approaches.
* The empirical results seem to demonstrate their method is competitive or better than other approaches.
* REBEL compares favourably in terms of runtime and memory usage with other, similarly performing methods.

Overall the theoretical and empirical examinations of their method seems very thorough.

**Weaknesses:**

See questions

**Questions:**

* Do the authors have any idea why REBEL seems to have a slightly higher KL than the other methods?
* Although in image alignment REBEL seems to do similarly to PPO, it also has higher variance. Do you know why that might be?
* Are the results for the 6.8B model significant? It seems as though REBEL produces very similar performance to e.g. PPO. For the smaller models the separation seems larger, is there a reason why the separation in performance between REBEL and other methods is bigger for smaller models?
* What are the error bars in Table 1? Is that standard deviation?

**Limitations:**

The authors discuss the limitations throughout their work at relevant stages.

---

> ### Author Rebuttal · Authors · 2024-08-07
>
> Thank you for your encouraging review and comments. We respond to your individual questions below.
>
> > Do the authors have any idea why REBEL seems to have a slightly higher KL than the other methods?
>
> The KL divergence is generally close across methods. For the TL;DR experiments, following previous work [1], we apply an addition KL penalty to the reward: $r(x, y) = RM(x, y) - \gamma (\ln \pi_{\theta_{t}} (y|x) - \ln \pi_{\theta_{0}} (y|x))$. PPO incorporates a clipping mechanism on top of this regularization, which allows it to control the KL-divergence more strictly. This difference in approach can lead to REBEL exhibiting a slightly higher KL-divergence compared to PPO.
>
> > Although in image alignment REBEL seems to do similarly to PPO, it also has higher variance. Do you know why that might be?
>
> To fairly compare REBEL and PPO, we tuned all hyperparameters, including parameters related to reward queries for PPO. Afterward, to ensure that REBEL was compared fairly to PPO, we kept those hyperparameters related to reward queries constant so that REBEL and PPO had the same number of reward queries per update. We believe that if we were to modify the reward queries per update, REBEL would show a lower variance than our current runs.
>
> > Are the results for the 6.8B model significant? It seems as though REBEL produces very similar performance to e.g. PPO. For the smaller models the separation seems larger, is there a reason why the separation in performance between REBEL and other methods is bigger for smaller models?
>
> The discrepancy in our training setup could be the reason for the smaller separation in the larger model versus the smaller models. Specifically, we performed hyperparameter tuning using LoRA [3] on the 1.4B and 2.8B models. We then directly applied these hyperparameters to the larger 6.9B model for full-parameter training without additional tuning (detailed in Appendix H.1.2). The results for the baselines that we compared to (REINFORCE, PPO, RLOO) for the 6.9B models are obtained from [1,2], which might have directly tuned the 6.9B model. These differences contribute to the decreasing performance gains as model size increases.
>
> > What are the error bars in Table 1? Is that standard deviation?
>
> The error bars are standard deviations. Results are averaged over three seeds and the standard deviations across seeds are in parentheses.
>
> [1] Huang S, Noukhovitch M, Hosseini A, Rasul K, Wang W, Tunstall L. The N+ Implementation Details of RLHF with PPO: A Case Study on TL; DR Summarization.
>
> [2] Ahmadian A, Cremer C, Gallé M, Fadaee M, Kreutzer J, Üstün A, Hooker S. Back to basics: Revisiting reinforce style optimization for learning from human feedback in llms.
>
> [3] Hu EJ, Shen Y, Wallis P, Allen-Zhu Z, Li Y, Wang S, Wang L, Chen W. Lora: Low-rank adaptation of large language models. arXiv

---

> > ### Comment · Reviewer_dEG5 · 2024-08-12
> > **Response**
> >
> > Thank you very much for the responses to my questions. I will maintain my score and continue to believe this is an excellent paper.

---

### Author Rebuttal · Authors · 2024-08-06

We thank all the reviewers for their time and insightful comments, which have significantly improved our paper. We are pleased that the reviewers appreciated our algorithm's simplicity, the detailed theoretical connections to prior methods, and the thorough empirical results. We summarize the main suggestions below and address each reviewer's comments individually in each reviewer’s rebuttal.

* Reviewer dEG5 raises questions regarding the experimental details and empirical results. We address these concerns in our detailed response to the reviewer.

* Reviewer HFsB provides interesting suggestions for exploring data aggregation and different choices of $\mu$. We would certainly explore this direction in future investigations into the REBEL algorithm.

* Reviewer ihUH expresses concerns about the empirical evidence supporting our claims. In response to this, we conduct an additional convergence experiment. We plot the reward vs. step for TL;DR dataset in the attached pdf. The plot demonstrates REBEL's faster convergence and higher rewards compared to iterative DPO and PPO.

* Reviewer zZ2j suggests including a separate “Limitation” section. We will certainly include this in the next version of the paper.

---

### Decision · Program_Chairs · 2024-09-25

**Decision:**

Accept (poster)

**Comment:**

The reviewers all agreed that this paper has clear presentation, strong theoretical contributions and good empirical evaluations. The simplicity of the approach can potentially make it very useful for practitioners, once tested at larger scales. This is a clear accept.